# 3D Molecular Generation via Virtual Dynamics

**Shuqi Lu**                                                                                        *lusq@dp.tech*
*DP Technology*

**Lin Yao**                                                                                         *yaol@dp.tech*
*DP Technology*

**Xi Chen**                                                                                        *chenx@dp.tech*
*DP Technology*

**Hang Zheng**                                                                                    *zhengh@dp.tech*
*DP Technology*

**Di He**                                                                                       *dihe@pku.edu.cn*
*Peking University*

**Guolin Ke**                                                                                       *kegl@dp.tech*
*DP Technology*

**Reviewed on OpenReview:** *https://openreview.net/forum?id=QvipGVdE6L*

## Abstract

Structure-based drug design, a critical aspect of drug discovery, aims to identify high-affinity molecules for target protein pockets. Traditional virtual screening methods, which involve exhaustive searches within large molecular databases, are inefficient and limited in discovering novel molecules. The pocket-based 3D molecular generation model offers a promising alternative by directly generating molecules with 3D structures and binding positions in the pocket. In this paper, we present VD-Gen, a novel pocket-based 3D molecular generation pipeline. VD-Gen features a series of carefully designed stages to generate fine-grained 3D molecules with binding positions in the pocket cavity end-to-end. Rather than directly generating or sampling atoms with 3D positions in the pocket, VD-Gen randomly initializes multiple virtual particles within the pocket and learns to iteratively move them to approximate the distribution of molecular atoms in 3D space. After the iterative movement, a 3D molecule is extracted and further refined through additional iterative movement, yielding a high-quality 3D molecule with a confidence score. Comprehensive experimental results on pocket-based molecular generation demonstrate that VD-Gen can generate novel 3D molecules that fill the target pocket cavity with high binding affinities, significantly outperforming previous baselines.

## 1 Introduction

Structure-based (pocket-based) drug design, i.e., finding a molecule to fill the cavity of the protein pocket with a high binding affinity (Kubinyi, 1993; DesJarlais et al., 1988; DeWitte et al., 1997; DeWitte & Shakhnovich, 1996), is one of the most critical tasks in drug discovery. The most widely used method is virtual screening (Walters et al., 1998; Shoichet, 2006; 2004). Virtual screening iteratively places molecules from a molecular database into the target pocket cavity and evaluates molecules with good binding based on rules such as energy estimation (de Ruiter & Oostenbrink, 2011; Chipot & Pohorille, 2007; Christ et al., 2010;

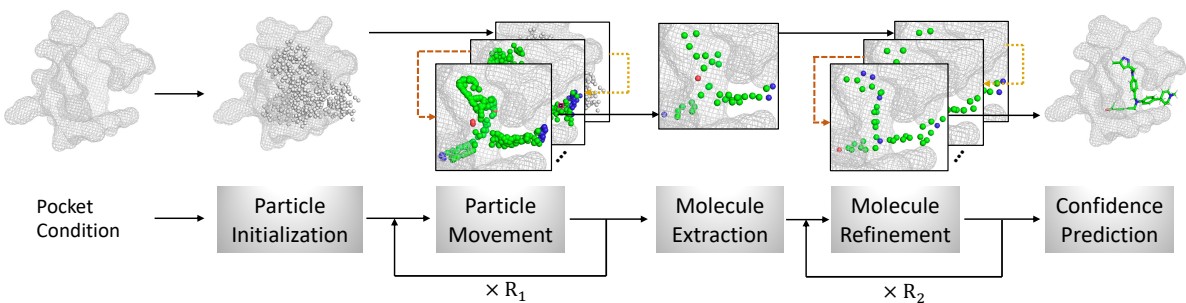

Figure 1: The `VD-Gen` pipeline. Given a pocket as the input, `VD-Gen` first initializes the Virtual Particles (VPs) and iteratively moves them to approximate the atom distribution in 3D space. Subsequently, a 3D molecule is extracted from the VPs after their movement. The atoms within the extracted 3D molecule are then further refined through additional iterations of movement. Lastly, a confidence score for the refined 3D molecule is predicted.

Michel & Essex, 2010). However, virtual screening is inefficient for the exhaustive search and is infeasible to generate new molecules that are not in the database. Recently, molecular generative models have become a potential solution to address the problem as they could generate novel molecules in an efficient way. The early attempts focused on ligand-based molecular generation (Kusner et al., 2017; Dai et al., 2018; Winter et al., 2019), which trains models to learn the underlying distribution of the molecules in training data and generate similar molecules. However, those methods did not consider conditional information, such as the shape of the pocket. Later, more efforts were paid to studying how to generate molecules conditioned on the information of protein pockets. Some pocket-based generative models simply generate molecules in the form of SMILES or graphs (Skalic et al., 2019; Xu et al., 2021), without considering the 3D geometric position of the molecule and pocket, which is closely related to binding affinity.

Given the 3D structure of a pocket, the ultimate goal of the task is to generate 3D molecules which contain a set of atoms, each with an atom type and the corresponding 3D position. Previous works can be broadly classified into two categories. 1) 3D density grid generation (Ragoza et al., 2022), in which pockets and molecules are converted to 3D density grids with coarse-grained positions, and then a generative model is used to predict the density at each grid. Since the model can only generate grid-level positions, these approaches cannot produce high-quality 3D molecules. 2) Auto-regressive 3D generation (Luo et al., 2022; Liu et al., 2022; Peng et al., 2022), in which atoms (with a 3D position and an atom type) are generated one by one. However, since it is hard to define the generation order of atoms during training, these models usually achieve inferior performance.

In this paper, we propose a new method, `VD-Gen`, which efficiently generates high-quality 3D molecules. The central concept of `VD-Gen` involves using a distribution of Virtual Particles (VPs) in 3D space to approximate the distribution of molecular atoms. As depicted in Fig. 1, the `VD-Gen` pipeline comprises five stages for end-to-end generation of 3D molecules: 1) Given a protein pocket, `VD-Gen` initializes multiple VPs. 2) The VPs are iteratively moved until equilibrium. Ideally, the distribution of equilibrated VPs closely approximates the distribution of molecular atoms. 3) A 3D molecule is extracted from the equilibrated VPs. 4) The atoms in the extracted molecule are refined through iterative movement. 5) A confidence score is predicted for the generated 3D molecule. By employing this pipeline, `VD-Gen` non-autoregressively generates high-quality 3D molecules and addresses issues in the previous works. In comparison to 3D grid-based generative models, `VD-Gen` produces high-quality 3D molecules with fine-grained coordinates. Furthermore, when contrasted with autoregressive generative models, `VD-Gen` efficiently generates all atoms simultaneously, resulting in improved performance that is not dependent on the generation order.

We have conducted comprehensive experiments utilizing various evaluation metrics, such as Vina (Trott & Olson, 2010), MM-PBSA (Yang et al., 2022), and 3D Similarity (Roy & Skolnick, 2015), in order to thoroughly benchmark `VD-Gen`. The experimental results indicate that our model is capable of generating diverse drug-like molecules exhibiting high binding affinities in 3D space, along with favorable binding poses, thereby significantly surpassing all baseline models. To further substantiate the effectiveness of `VD-Gen`, we

have designed ablation studies, case studies, and visualizations. Moreover, the `VD-Gen` pipeline can be readily extended to pocket-based 3D molecular optimization, achieving exceptional performance in this task as well.

## 2    Method

The aim of structure-based molecular generation is to learn a probabilistic mapping $f : \mathcal{P} \to \mathcal{D}(\mathcal{M})$, where $\mathcal{P}$ is the space of protein pockets, $\mathcal{M}$ is the space of all molecules, and $\mathcal{D}(\mathcal{M})$ is the probability distributions over molecules. Within this framework, a protein pocket, denoted as $\mathbf{P} \in \mathcal{P}$, is described by a set of $u$ atoms $\{(\boldsymbol{x}_i^p, \boldsymbol{y}_i^p)\}_{i=1}^u$, where each atom is represented by a one-hot encoded vector $\boldsymbol{x}_i^p \in \mathbb{R}^t$ that identifies its atomic type among $t$ possible types, and a vector $\boldsymbol{y}_i^p \in \mathbb{R}^3$ specifying its 3D spatial coordinates. For a given protein pocket $\mathbf{P}$, the objective is to produce a probability distribution over molecules, which assigns higher probabilities to molecules likely to bind to $\mathbf{P}$. This is achieved through a generative model $h(\mathbf{P}; \boldsymbol{\theta})$, where $h(\cdot; \boldsymbol{\theta})$ is a function modeling the generative process, parameterized by $\boldsymbol{\theta}$. This model accepts a specific pocket $\mathbf{P}$ as input and outputs a corresponding distribution over $\mathcal{M}$, where we can sample a molecule instance, denoted as $\mathbf{M}$. The sampled molecule $\mathbf{M}$ is defined as a set of $m$ atoms, $\mathbf{M} = \{(\boldsymbol{x}_i, \boldsymbol{y}_i)\}_{i=1}^m$.

Given the challenges associated with directly generating the distribution, we prefer an iterative refinement approach that begins with random initialization. Inspired by Molecular Dynamics, we initially randomize both the types and coordinates of atoms. These atoms are then iteratively refined by a model to approximate their ground-truth positions. For the success of this methodology, it is crucial to map each input atom to a unique target within the ground truth atoms, ideally by minimizing displacement.That is, we can randomly initialize the same number of atoms as in the ground-truth molecule and use the Wasserstein distance to assign a unique training target for each atom. While this approach is straightforward, we observed unstable training (i.e., divergence) and inferior performance. Further details are delineated in Appendix B.

To avoid training instability, we introduce Virtual Particles (VPs) and use many VPs (several times the atom number) to approximate the atom distribution of the ground-truth molecule in 3D space. In particular, a set of VPs is randomly allocated within the pocket cavity, serving as a distribution in 3D space. Subsequently, the VPs are gradually moved based on learnable dynamics, approximating the ground truth molecular atoms. We refer to this iterative process as `Virtual Dynamics`. As this dynamics is easily learnable, the training is stable, effectively addressing the unstable training issue in our initial attempt.

To extract valid molecules from the distribution formed by VPs, we design a filter-then-merge approach and subsequently refine the atoms in the extracted molecule using iterative movement. Moreover, considering real-world applications, `VD-Gen` predicts confidence scores for the generated 3D molecules. We describe the generation pipeline of `VD-Gen` in the following subsections and Algorithm 3.

### 2.1    VD-Gen Pipeline

**Particle Initialization**    The objective of this stage is to initialize VPs within the protein pocket cavity. First, the cavity space must be detected, where the VPs will be initialized. Given the atoms of a pocket, we use a 3D U-Net model (Çiçek et al., 2016) to predict the grids of cavity space. It is important to note that this 3D U-Net is trained independently, rather than in an end-to-end manner with the VD-Gen pipeline, as it solely detects the cavity space. The details of this 3D U-Net can be found in Appendix A.2. Next, we need to determine the number of initialized VPs. A neural network model is used to predict the number of atoms in the generated molecule based on pocket atoms, denoted as $\bar{m} = h_{an}(\mathbf{P}; \boldsymbol{\theta}_{an})$, where $h_{an}$ is a neural model with learnable parameter $\boldsymbol{\theta}_{an}$ and $\bar{m}$ is the predicted atom count. The number of VPs is then set as $n = \bar{m} k_{vp}$, where $k_{vp} \in \mathbb{N}$ is a hyperparameter. Finally, we randomly and uniformly distribute $n$ VPs within the predicted cavity space and denote the initialized VPs as $\mathbf{V}_0$. It is worth emphasizing that we utilize $k_{vp}$ times the predicted atom count to ensure comprehensive VP coverage within the cavity. Sufficient coverage permits the nearest assignment approach for VPs, subsequently minimizing the learning complexity.

**Particle Movement**    The objective of this stage is to iteratively move the initially randomized VPs with their respective ground-truth atoms.Given an initial set of VPs, $\mathbf{V}_0$, we update their distribution by moving them in 3D space, to approximate the ground-truth atom distribution. This movement process is iterative, similar to Molecular Dynamics. Specifically, at each iteration, the model takes the positions and types of

VPs from the previous iteration as inputs and produces updated positions and types. This process can be represented as $\mathbf{V}_{r+1} = f_{pm}(\mathbf{V}_r, \mathbf{P}; \boldsymbol{\theta}_{pm})$, where $\mathbf{V}_r = \{(\boldsymbol{x}_i^r, \boldsymbol{y}_i^r)\}_{i=1}^n$ denotes the set of $n$ VPs predicted at the $r$-th iteration, $f_{pm}$ is an SE(3) equivariant model that accepts 3D coordinates as inputs, and $\boldsymbol{\theta}_{pm}$ are the learnable parameters. The number of iterations in the *Particle Movement* stage is denoted by $R_1$.

**Molecule Extraction** The objective of this stage is to extract molecules from $\mathbf{V}_{R_1}$, by merging VPs. Given that VPs gravitate towards ground-truth atoms, positions with denser VP concentrations are probable atomic locations. Consequently, the merging strategy primarily leverages the 3D coordinates of the VPs, allowing us to select the top-$m$ densest locations for molecular extraction. To achieve this, we have designed a neural network-based model.Formally, this model can be represented as $\mathbf{W}_0 = h_{me}(\mathbf{V}_{R_1}, \mathbf{P}; \boldsymbol{\theta}_{me})$, where $\boldsymbol{\theta}_{me}$ denotes learnable parameters, and $\mathbf{W}_0 = \{(\hat{\boldsymbol{x}}_i^0, \hat{\boldsymbol{y}}_i^0)\}_{i=1}^m$ is the set of $m$ atoms in the extracted 3D molecule. The model constructs $m$ atoms from $n$ VPs through a two-step process: filtering and merging. Both these steps predominantly utilize the atomic coordinates with the intention of first eliminating VPs with significant errors and then merging VPs to produce stable representative atoms.

First, some VPs might not align precisely to their targeted positions; our goal is to eliminate such outliers.To achieve this, we predict the errors (distances between VPs and their target positions) for VPs, and exclude those with significant errors. Specifically, errors surpassing $\zeta$ (a predetermined hyper-parameter) are deemed significant.

Second, we merge the filtered VPs into atoms. For every pair of VPs, the model predicts a merging probability. With the predicted pairwise merging probability matrix, we establish a threshold, resulting in a binary merging matrix. VPs are then grouped into clusters based on this matrix. However, determining an appropriate threshold is challenging. Notably, we have already predicted the number of atoms, $\bar{m}$, during the *Particle Initialization* stage. As such, We employ a binary search to determine a merging threshold that results in a cluster count approximating $\bar{m}$.

We then obtain several (ideally $\bar{m}$) merged clusters, denoted as $\boldsymbol{w}_i$ for the set of indices of the $i$-th cluster's VPs. To compute $\mathbf{W}_0 = \{(\hat{\boldsymbol{x}}_i^0, \hat{\boldsymbol{y}}_i^0)\}_{i=1}^m$, within each cluster, we use $\hat{\boldsymbol{x}}_i^0 \sim \text{Uniform}(\{\boldsymbol{x}_j^{R_1} | j \in \boldsymbol{w}_i\})$ and $\hat{\boldsymbol{y}}_i^0 = \text{Mean}(\{\boldsymbol{y}_j^{R_1} | j \in \boldsymbol{w}_i\})$ to sample an atom type and obtain an average coordinate, respectively. It's essential to note that while the average coordinate of the VPs in a cluster offers a stable representative position, atom types are discrete. Therefore, determining an average type is not possible, which leads us to select a type from the VPs within the cluster. Notably, we refer to the particles in $\mathbf{W}_0$ as atoms to distinguish them from VPs in $\mathbf{V}_0$.

**Molecule Refinement** The objective of this stage is to further refined the atoms in $\mathbf{W}_0$, to obtain a high-quality 3D molecule with fine-grained 3D coordinates.This stage is similar to the *Particle Movement*, albeit with different input (the atoms in $\mathbf{W}_0$) and different model parameters. Formally, the process in this stage can be represented as $\mathbf{W}_{r+1} = f_{mr}(\mathbf{W}_r, \mathbf{P}; \boldsymbol{\theta}_{mr})$, where $f_{mr}$ denotes the SE(3) equivariant model with learnable parameter $\boldsymbol{\theta}_{mr}$. The term $\mathbf{W}_r = \{(\hat{\boldsymbol{x}}_i^r, \hat{\boldsymbol{y}}_i^r)\}_{i=1}^m$ represents the set of $m$ atoms at the $r$-th iteration. The total number of iterations in the *Molecule Refinement* process is denoted by $R_2$.

**Confidence Prediction** In practical applications, it is often necessary to generate multiple molecules and select the top ones based on their binding affinities. Consequently, a reliable confidence predictor is essential for ranking these molecules. While computational simulations and experimental methods can be used to evaluate the generated molecules, these approaches are often prohibitively expensive, particularly when dealing with a large number of candidates. To enhance the usability of the proposed method and minimize additional costs associated with molecule selection, this stage aims to predict a confidence score for each generated 3D molecule. Specifically, we adopt the approach from AlphaFold2 (Jumper et al., 2021), utilizing the pLDDT (predicted LDDT) score as our confidence metric, enabling the ranking and selection of the most promising molecules.

## 2.2 VD-Gen Training Strategies

In the VD-Gen training, we first train the 3D U-Net used for pocket cavity detection independently (refer to Appendix A.2 for details). Then, we conduct end-to-end training for the entire VD-Gen pipeline, based on the

3D U-Net predicted pocket cavity. The following subsections provide a detailed description of the training strategies implemented at each stage.

**Training of Particle Movement**   The objective of the model $f_{pm}$ is to move the virtual points (VPs) to the positions of the ligand molecular atoms, thereby approximating the distribution of molecular atoms. To accomplish this, a real atom can be directly assigned as the training target for each VP.

Formally, given the ground-truth atoms $\mathbf{G} = \{(\boldsymbol{x}_i^g, \boldsymbol{y}_i^g)\}_{i=1}^m$ and the initialized VPs $\mathbf{V}_0$, there are $n^m$ possible assignments. In accordance with the principle of least action (Feyman, 1965), the assignment with the minimal moving distance is preferred. This entails optimizing $\text{Min} \sum_{i=1}^n \|\boldsymbol{y}_i^0 - \boldsymbol{y}_{a_i}^g\|_2$, where $\boldsymbol{y}_i^0$ denotes the initial position and $a_i \in \mathbb{N}$ represents the assigned target for the $i$-th VP. This optimization problem can be easily solved by assigning the nearest real atom as the training target for the $i$-th VP, i.e., $a_i = \arg\min_{j=1}^m \|\boldsymbol{y}_i^0 - \boldsymbol{y}_j^g\|_2$. However, this nearest assignment may result in some real atoms not being assigned as training targets. To increase coverage, there are two approaches. The first involves considering atom coverage as a constraint in the aforementioned optimization problem, while the second entails using more VPs to cover the pocket cavity as extensively as possible. Although the former is more favorable from an algorithmic perspective, it increases the learning difficulty of VP movement, as the constrained assignment violates the principle of least action. Therefore, we adopt the latter approach, using more VPs to increase coverage.

Given the assigned targets $a_i$, we use the following losses for the training. First, a negative log-likelihood loss is used for the VPs' types. Second, a clip L2 loss is used for the VPs' 3D coordinates. Finally, a regularization loss is used to limit the moving distances between two adjacent iterations, to stabilize the training. Combined above, the final training loss function at the $r$-th iteration could be denoted as

$$\mathcal{L}_{pm} = \frac{1}{n} \sum_{i=1}^n \left( \text{NLL}(\bar{\boldsymbol{x}}_i^r, \boldsymbol{x}_{a_i}^g) + \text{clip}(\|\boldsymbol{y}_i^r - \boldsymbol{y}_{a_i}^g\|_2, \tau) + \max(\|\boldsymbol{y}_i^r - \boldsymbol{y}_i^{r-1}\|_2 - \delta, 0) \right), \tag{1}$$

where $\bar{\boldsymbol{x}}_i^r$ is the predicted vector of atom types of $i$-th VP, $\boldsymbol{x}_{a_i}^g$ is the ground-truth atom types of $i$-th VP, $\boldsymbol{y}_i^r$ ($\boldsymbol{y}_{a_i}^g$) is the predicted (ground-truth) coordinates of $i$-th VP, $\tau$ is the the clip value for coordinate loss, and $\delta$ is the threshold for moving regularization.

It is important to note that there are $R_1$ iterations in this stage. However, training the model using multiple iterations is inefficient in terms of both computational speed and memory consumption. To reduce the training cost, we adopt the stochastic iteration approach used in AlphaFold2 (Jumper et al., 2021). Specifically, during the training process, the iteration $r$ is uniformly sampled between 1 and $R$, where $R$ represents the maximum iteration ($R = R_1$ in this stage). The model is then executed in forward-only mode for the first $r-1$ iterations, without calculating the loss or performing gradient backpropagation. The gradient calculation and backpropagation are enabled only at the $r$-th iteration. During the inference phase, this sampling of iterations is not employed. The aforementioned algorithm is illustrated in Algorithm 1.

**Training of Molecule Extraction**   In this stage, there are two training tasks. The first task is to predict the errors, which are the distances between the VPs and their target positions. To stabilize the training, we bucket the errors into one-hot bins and convert the task into a classification problem, following the pLDDT training in AlphaFold2 (Jumper et al., 2021):

$$\mathcal{L}_{error\_pred} = \frac{1}{n} \sum_{i=1}^n \text{NLL}(\bar{\boldsymbol{s}}_i, \boldsymbol{s}_i), \tag{2}$$

where $\boldsymbol{s}_i$ represents the one-hot vector of the bucketed target error bin, and $\bar{\boldsymbol{s}}_i$ denotes the predicted probability vector.

The second task aims to predict which VP pairs should be merged. Ideally, VPs with the same target atom should be merged; therefore, the training label for a VP pair with the same target atom is set to "true". Given $n$ VPs and $m$ real atoms, the ratio of the "true" class is approximately $\frac{m \times (n/m)^2}{n^2} = \frac{1}{m}$. As $m$ ranges from dozens to hundreds, this binary classification task is highly imbalanced. To address this issue, we introduce a

---

**Algorithm 1** Iterative Movement

---

**Require:** $R$: max iterations, **P**: pocket atoms, $\mathbf{V}_0$: random initialized VPs, $f(\cdot; \boldsymbol{\theta})$: SE(3) equivariant model with parameters $\boldsymbol{\theta}$

1: $r \leftarrow$ uniform(1, $R$) if training else $R$            ▷ *Sampling is only enabled at training*
2: disable_gradient()            ▷ *Disable gradient calculation globally*
3: **for** $k \in [1, ..., r-1]$ **do**
4:      $\mathbf{V}_k \leftarrow f(\mathbf{V}_{k-1}, \mathbf{P}; \boldsymbol{\theta})$            ▷ *update without gradients*
5: enable_gradient()            ▷ *Enable gradient calculation globally*
6: $\mathbf{V}_r \leftarrow f(\mathbf{V}_{r-1}, \mathbf{P}; \boldsymbol{\theta})$            ▷ *update with gradients*
7: **return** $\mathbf{V}_r$            ▷ *Return the positions and types of particles*

---

focal loss (Lin et al., 2017) to balance the classes.

$$\mathcal{L}_{merge} = \frac{1}{l^2} \sum_{i=1}^{l} \sum_{j=1}^{l} FL(\bar{\boldsymbol{r}}_{ij}, \boldsymbol{r}_{ij}), \tag{3}$$

where $l$ denotes the number of VPs after filtering, $\boldsymbol{r}_{ij}$ represents the target merging type, and $\bar{\boldsymbol{r}}_{ij}$ indicates the predicted probability of merging type.

**Training of Molecule Refinement** The objective of the model $f_{mr}$ is to refine the atoms in $\mathbf{W}_0$ to ligand molecular atoms. Similar to the *Particle Movement* stage, a real atom can be directly assigned as the training target for each atom. Since $\mathbf{W}_0$ is constructed from the clusters of VPs, the training target for each atom is determined by its corresponding cluster of VPs. To be specific, we first collect all the training targets of VPs belonging to a particular cluster, and then select the most frequently occurring one. Formally, for the $i$-th atom, its training target atom is denoted as $b_i = \text{most\_frequent}(\{a_j | j \in \boldsymbol{w}_i\})$, where $\boldsymbol{w}_i$ represents the indices of VPs belonging to the $i$-th cluster and $a_j$ is the target atom for the $j$-th VP. Subsequently, the same training loss functions employed in the *Particle Movement* stage can be used to train the model $f_{mr}$.

**Training of Confidence Prediction** We explicitly train a task to learn the confidence scores for generated molecules. In particular, following the approach in AlphaFold2 (Jumper et al., 2021), we compute the Local Distance Difference Test (LDDT) score (Mariani et al., 2013) for both the generated and ground-truth molecules, employing a model to predict the LDDT score. Furthermore, we bucket the LDDT score into one-hot bins, effectively transforming it into a classification task, mirroring the approach in AlphaFold2 (Jumper et al., 2021).

$$\mathcal{L}_{confidence} = \frac{1}{n} \sum_{i=1}^{n} \text{NLL}(\bar{\boldsymbol{e}}_i, \boldsymbol{e}_i), \tag{4}$$

where $\bar{\boldsymbol{e}}_i$ is the predicted LDDT (pLDDT) probability distribution, and $\boldsymbol{e}_i$ is the one-hot vector of the bucketed LDDT bins. Through the extraction of the expected value from the pLDDT probability distribution, a scalar pLDDT value can be derived, providing a confidence score to rank and prioritize molecules.

**SE(3) Equivariant Model** Both $f_{pm}$ and $f_{mr}$ necessitate the use of SE(3) equivariant models that accept 3D coordinates as inputs and yield new 3D coordinates as outputs. We mainly follow the design of the efficient SE(3)-equivariant Transformer proposed in Uni-Mol (Zhou et al., 2023) and Graphormer-3D (Shi et al., 2022). However, these models do not account for the interaction between the pocket and the molecule. As a result, we extend the model with an additional pocket encoder and incorporate a particle-pocket attention mechanism to capture the interactions between the pocket and the molecule. Since the focus of this paper is on pocket-based 3D molecular generation rather than the SE(3) equivariant models themselves, we provide the details of the designed SE(3) equivariant model in Appendix A.1.

## 2.3 Extending VD-Gen to 3D Molecular Optimization

Molecular optimization is a crucial aspect of practical drug design. This process involves refining an existing molecule by replacing a fragment to improve its binding affinity, rather than creating a new molecule from scratch. Here, we have extended the VD-Gen method to encompass pocket-based 3D molecular optimization.

As depicted in Fig. 8, the procedure commences with the random removal of a fragment from the given molecule. The model is then trained to regenerate this fragment, using the pocket and remaining atoms of the molecule as conditional inputs. Although the model is not explicitly trained for molecular optimization, it acquires the ability to remove and replace fragments within a molecule, thereby rendering it applicable to optimization tasks. Detailed results of the molecular optimization benchmarks can be found in Appendix C.6.

## 3 Experiments

### 3.1 Settings

**Evaluation Metrics** There is no single golden metric for evaluating generated molecules; thus, we employ multiple metrics to ensure a comprehensive assessment.

1) *3D Similarity.* Since pocket-based 3D generation models are trained using 3D structures of pockets and molecules, the most direct metric for assessing the models' generative capabilities is to evaluate the 3D similarity between the generated molecule and the ground-truth one. We use LIGSIFT (Roy & Skolnick, 2015) to calculate the overlapping ratio in 3D space between two molecules.

2) *Vina.* Docking scores, such as Vina (Trott & Olson, 2010), are commonly used in previous pocket-based generation studies due to their ease of computation. To maintain consistency with prior works, we also use **Vina** as a metric. However, previous studies often relied on Vina's re-docking, which could significantly alter molecular conformation and binding pose due to docking tools. Therefore, to directly assess the 3D molecules generated by our model, we introduce an additional **Vina\*** score that does not use re-docking.

3) *MM-PBSA.* Although docking scores are quick and easy to compute, they were designed to identify potential hits in large-scale virtual screening, not for ranking. Consequently, docking scores are not suitable metrics for comparing binding affinities across different models (Cheng et al., 2009). As a result, we employ the slower but more accurate MM-PBSA (Molecular Mechanics Poisson–Boltzmann Surface Area) (Genheden & Ryde, 2015) metric. Based on MM-PBSA, we introduce two additional metrics: **MM-PBSA B.T.** (MM-PBSA Better than Target), which calculates the percentage of generated molecules with better MM-PBSA scores than the ground-truth, and **MM-PBSA Rank**, which computes the average rankings of different models among various complexes. Since MM-PBSA scores can vary significantly between complexes, MM-PBSA Rank offers a more effective comparison of different models.

The details of these metrics are described in Appendix C.2.

**Data** we use the same training dataset as used in previous works (Luo et al., 2022; Peng et al., 2022), specifically the CrossDocked data (Francoeur et al., 2020), to train `VD-Gen`. The CrossDocked dataset originally contains 22.5 million docked protein-ligand pairs. Luo et al. (2022) filter out data points whose binding pose RMSD is greater than 1Å, and use MMSeqs2 (Steinegger & Söding, 2017) to cluster the data, and randomly select 100,000 protein-ligand pairs for training. For the test set, we utilize 100 protein-ligand complex crystal structures from (Yang et al., 2022), on which MM-PBSA was validated to be effective. To assess the diversity of our test set, we employed MMSeqs2 to cluster the protein sequences, maintaining a threshold of 40% sequence identity. Our analysis yielded approximately 73 distinct clusters, indicating significant diversity within the test set. To prevent data leakage, we exclude complexes from the training data whose protein sequences bear similarity to those in the test set. Specifically, we consider two protein sequences as similar if their e-value, obtained from the BLAST search results (Camacho et al., 2009), is greater than 0.4.

**Training** Initially, the 3D U-Net model used for the pocket cavity detection is trained independently, requiring around 20 hours on 8 NVIDIA A100 GPUs. Subsequently, the parameters of the 3D U-Net model are frozen, and the entire `VD-Gen` pipeline is trained end-to-end, taking approximately 15 hours on 8 NVIDIA A100 GPUs. Detailed hyper-parameters used during training can be found in Appendix C.1.

### 3.2 Molecule Generation Performance

**Baselines** We compare the `VD-Gen` with several previously proposed 3D pocket-based molecular generation models, including the 3D density grid generative model LiGAN (Ragoza et al., 2022), the auto-regressive 3D

Table 1: Performance on pocket-based 3D molecular generation. We additionally computed the standard deviations for VD-Gen's results.

| Model | 3D Sim($\uparrow$) | Vina($\downarrow$) | Vina*($\downarrow$) | MM-PBSA($\downarrow$) | MM-PBSA-Rank($\downarrow$) | MM-PBSA-B.T.(%$\uparrow$) |
|---|---|---|---|---|---|---|
| LiGAN (Ragoza et al., 2022) | 0.356 | -6.724 | -5.372 | -18.462 | 2.59 | 0.3 |
| 3DSBDD (Luo et al., 2022) | 0.365 | -8.662 | -7.227 | -30.560 | 2.31 | 2.29 |
| GraphBP (Liu et al., 2022) | 0.333 | -8.710 | -3.689 | -5.579 | 4.01 | 0 |
| Pocket2Mol (Peng et al., 2022) | 0.352 | -8.332 | -6.525 | -8.226 | 3.53 | 0 |
| VD-Gen | **0.422** | **-8.998** | **-7.404** | **-50.744** | **1.16** | **11.623** |
| | $\pm0.0001$ | $\pm0.0009$ | $\pm0.0257$ | $\pm0.138$ | | $\pm0.163$ |

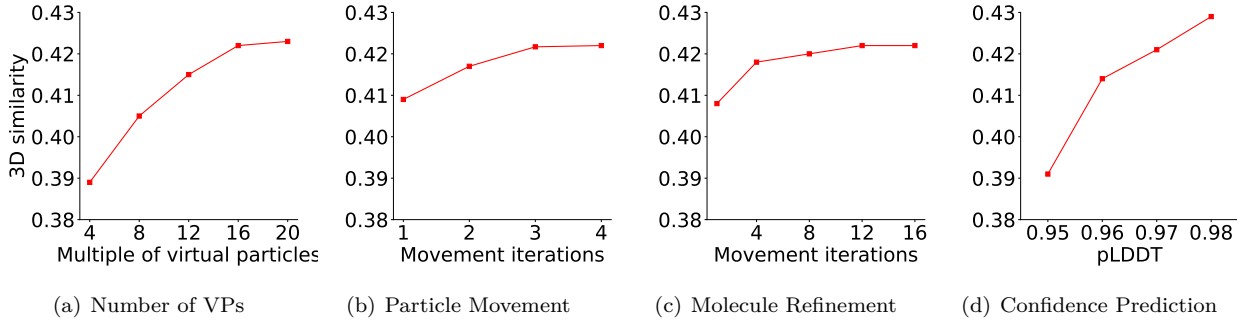

(a) Number of VPs  (b) Particle Movement  (c) Molecule Refinement  (d) Confidence Prediction

Figure 2: Ablation studies for VD-Gen pipeline.

generative models GraphBP (Liu et al., 2022), 3DSBDD (Luo et al., 2022), and Pocket2Mol (Peng et al., 2022). For each model, we generate 500 molecules for every pocket and subsequently select 100 of these molecules for evaluation. In the cases of 3DSBDD and Pocket2Mol, beam search is used and the top 100 molecules are chosen. For VD-Gen, the selection is based on the confidence score. For LiGAN and GraphBP, 100 molecules are randomly selected, as these models do not implement beam search. Additionally, We trained three VD-Gen models with different seeds, computed the mean and standard deviation of the inference results for VD-Gen.

**Results**   As we pay more attention to the generated molecules with high binding affinities, we report the top 5-th percentile result for Vina, Vina*, and MM-PBSA. MM-PBSA-Rank is calculated based on the top 5-th percentile MM-PBSA result. The 10-th, 25-th, and 50-th percentile results are in Appendix C.3.

From the results in Table 1, it is easy to conclude: 1) VD-Gen significantly outperforms all other baselines in all metrics, with top-1 MM-PBSA Rank, demonstrating the superior performance of the proposed VD-Gen. 2) MM-PBSA B.T shows that VD-Gen can generate more molecules with better MM-PBSA scores than the ground-truth ones, while baseline hardly can. 3) In 3D Similarity results, VD-Gen also largely outperforms baselines, indicating that VD-Gen effectively learned the pocket-based 3D molecular generation and can generalize to unseen pockets. 4) Although some baselines achieve good performance on Vina scores, like GraphBP and Pocket2Mol, their Vina* and MM-PBSA scores are very poor. We believe the re-docking in Vina fixes their generated 3D structures and then a good Vina score could be obtained. This result indicates that the previously widely used Vina score is not a good metric for pocket-based 3D molecular generation.

To summarize, the superior results on multiple evaluation metrics explicitly demonstrate the effectiveness of the proposed VD-Gen.

### 3.3  Ablation Study

**Number of VPs**   VPs are employed to approximate the distribution of molecular atoms. The accuracy of this approximation is intuitively expected to improve with an increased number of VPs. Therefore, we investigate the impact of the number of VPs on the overall performance, with the findings presented in

Fig. 2(a). The results clearly demonstrate that performance is influenced by the number of VPs, with higher numbers yielding better performance. However, we observe that the performance stabilizes when the number of VPs reaches 16 times the predicted molecular atoms. This suggests that utilizing an excessive number of VPs is unnecessary, as an appropriate balance between efficiency and performance can be achieved with a more moderate quantity.

**Number of Movement Iterations**  Iterative movement is critical in the `VD-Gen`. In Fig. 2(b) and Fig. 2(c), we benchmark the effectiveness of different iterations in *Particle Movement* and *Molecule Refinement*. For the results in Fig. 2(b), we reduce the iterations $R_2$ to $0.25R_2$ in *Molecule Refinement* stage, to better show the gain brought by *Particle Movement* stage. As shown in Fig. 2(b) and Fig. 2(c), we can find more iteration iterations improve the final performance in both two stages.

**Effectiveness of Molecule Refinement**  The *Molecule Refinement* stage is used to further refine the 3D molecule extracted by *Molecule Extraction*. To examine how *Molecule Refinement* affects the final performance, we benchmarked different iterations. As shown in Fig. 2(c), we can find the results with more iterations are better. The result indicates the necessity of the *Molecule Refinement* stage.

**Effectiveness of Confidence Prediction**  The pLDDT score, outputed at *Confidence Prediction*, is utilized for selecting and ranking molecules, and we aim to evaluate its effectiveness. Specifically, we compute the correlation between 3D similarity and the pLDDT for the generated molecules within a pocket (PDBID 1LF2), with the results displayed in Fig. 2(d). It is evident that a higher pLDDT score corresponds to better 3D similarity. This finding suggests that the confidence score provided by `VD-Gen` is a reliable indicator for selecting and ranking the generated molecules.

### 3.4  Case Study

Here, we selected 5 protein pockets from the test set to visualize the results generated by `VD-Gen` for pocket-based generation tasks. Fig. 3 displays three top-scoring molecules (purple, middle column) for each pocket, based on MM-PBSA scores. These molecules are presented without any structural post-processing. Green molecules represent ground truth molecules, and the rightmost column illustrates the spatial overlap between the generated and original molecules.

In the first case (PDBID: 2XBW), the protein pocket features a deep pit within the protein (bottom left of the image) that can accommodate approximately one benzene ring. This task is challenging due to the small size of the pit and its considerable distance from the pocket's center. The results reveal that VD-Gen successfully generated molecules with fragments grown within the pit. While the generated molecules exhibit good 3D similarity to the original molecules and favorable MM-PBSA scores, the Vina scores of the original molecule are substantially better than those of the generated molecules. Relying solely on Vina scores for molecule selection may result in the exclusion of optimal candidates.

In the second case (PDBID: 1BHX), the protein pocket is characterized by a bulky structure, necessitating the generation of protein-interacting fragments at both ends of the pocket and connecting them via a molecular backbone. The original molecule is elongated and distorted, rendering this prediction task challenging. VD-Gen-generated molecules effectively replicate the shape of the original molecules, adequately filling the irregular protein pockets. All three molecules exhibit good 3D similarity and MM-PBSA scores.

In the third case (PDBID: 2BRM), the protein pocket is flat, which requires that the molecular backbone of the ligand bound to it should be close to a planar structure, such as composed of conjugated aromatic rings. We can see that the molecules generated by `VD-Gen` are the same as the original molecular structures, whose molecular backbone is a planar structure composed of conjugated aromatic rings, and the part toward the outside of the pocket is flexible. We can see that in this case, Vina scoring, MM-PBSA scoring, and 3D similarity all show good agreements.

In the fourth case study (PDBID: 1D3P), the pocket presents a Y-shaped configuration, with the native ligand forming a branched structure from the protein's interior to the exterior. The entire molecular structure is rather elongated, which adds to the complexity of the generation process. The molecules generated by

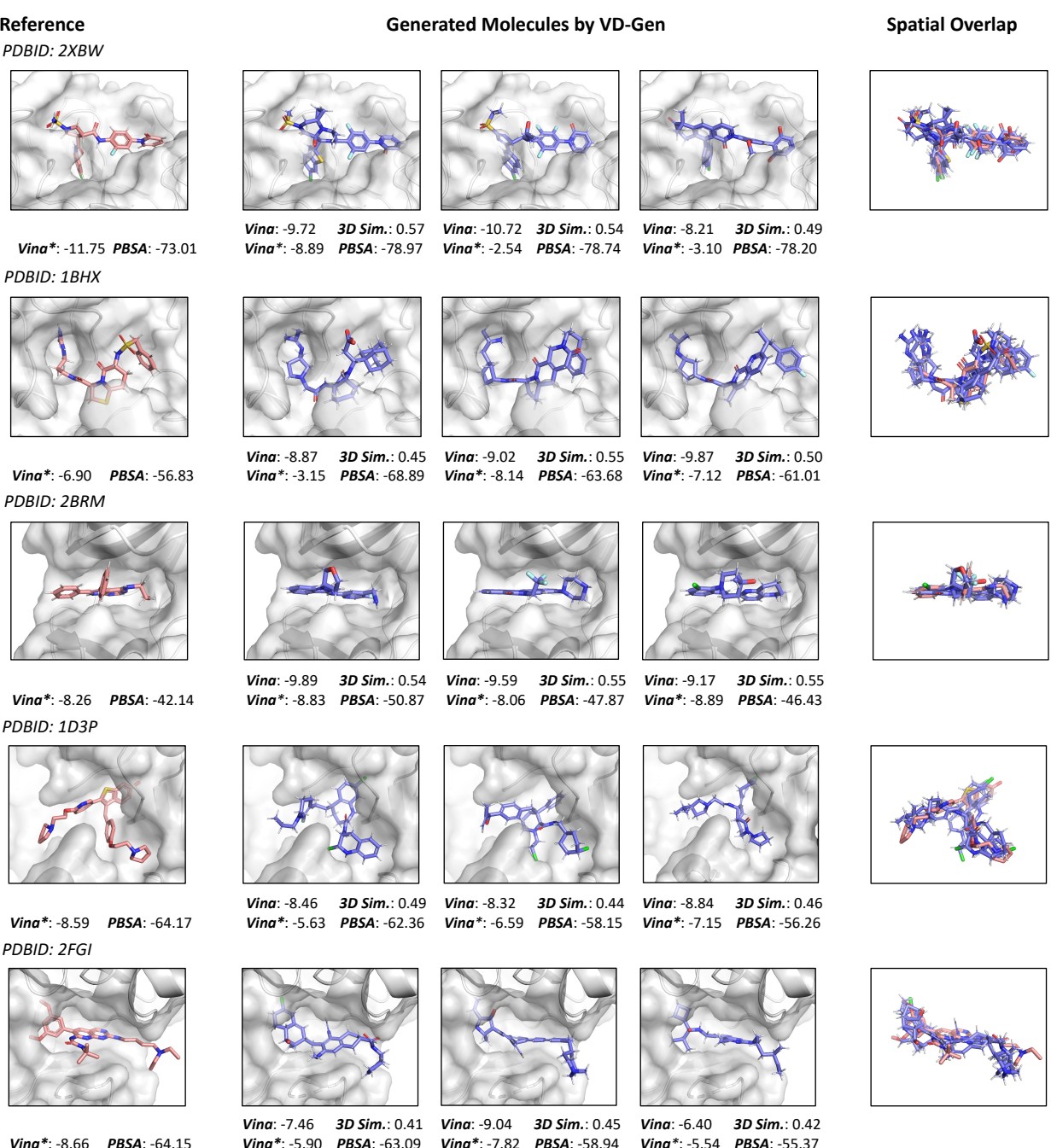

Figure 3: Generated molecules with high 3D similarity to the reference molecule and high PBSA scores for five protein pockets. Gray surfaces are the protein pockets. Pink molecules are the ground truth molecules. Purple molecules are the molecules generated by VD-Gen. Lower Vina score, lower PBSA score and higher 3D similarity indicate higher binding affinity.

VD-Gen successfully reproduce this branched shape, and the length of each branch is nearly identical to that of the original ligand.

In the fifth case study (PDBID: 2FGI), a portion of the original ligand is oriented towards the solvent environment (on the right side of the image). This segment does not strictly require specific structural

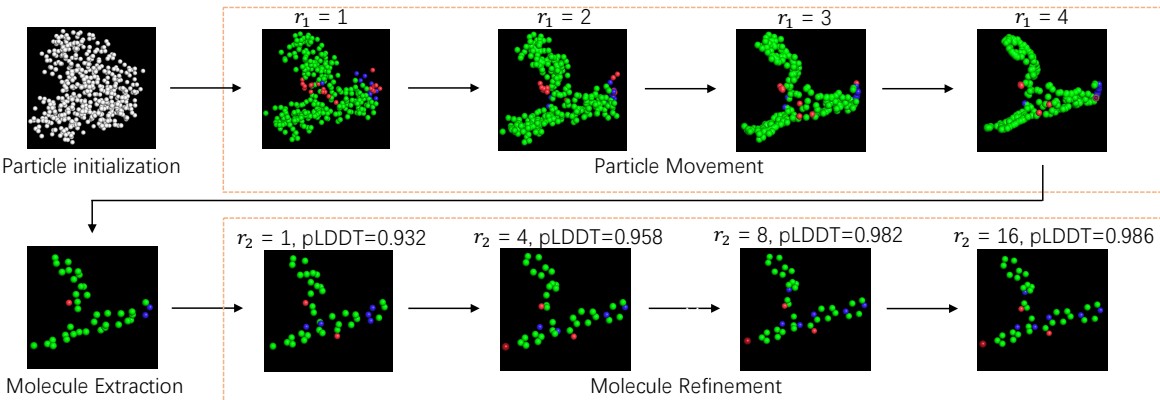

Figure 4: An example to demonstrate the output results of `VD-Gen`'s different stages, $r_1$ ($r_2$) is the iteration number of *Particle Movement* (*Molecule Refinement*). The increasing pLDDT scores in the pipeline indicate the effectiveness of `VD-Gen`.

features, as it does not interact with the protein. The molecules generated by `VD-Gen` achieve good overlay with the original ligand in the pocket's interior, while adopting diverse structures near the solvent. This indicates that VD-Gen can effectively identify pockets and produce ligands with specificity. Consequently, the 3D similarity in this group may be slightly lower.

The cases presented in Figure 3 demonstrate `VD-Gen`'s robust generation capabilities across various challenging molecular generation tasks. The generated molecules can fill deep pockets, conform to large pocket trends, match specific pocket structures and adopt diverse structures in the non-binding parts (such as near the solvent),and they show high 3D similarity to the molecules in the original crystal structure. Additionally, the MM-PBSA scores and 3D similarity consistently evaluate the quality of the generated molecules, whereas Vina scores are less reliable in some instances. This finding suggests that selecting molecules based solely on Vina scores may be inadequate.

### 3.5 Visualization

To better understand how `VD-Gen` generates the 3D molecules, we present a visualization for the various stages in `VD-Gen`, as illustrated in Fig. 4. Initially, the VPs are uniformly distributed within the protein pocket. During the *Particle Movement* stage, with more iterations, the VPs gradually aggregate into several clusters, converging towards the positions of molecular atoms. Subsequently, in the *Molecule Extraction* stage, a 3D molecule with fewer atoms is extracted. Finally, the *Molecule Refinement* stage involves further refinement of the extracted 3D molecule to achieve more accurate 3D positions with improved pLDDT scores.

## 4 Related Work

**Ligand-Based Molecular Generation** Early studies on ligand-based molecular generation utilized sets of molecules as training data, generating new molecules based on the learned distribution. These methods primarily represented molecules as 1D SMILES strings and 2D molecular graphs, employing techniques such as variational autoencoders (VAEs) (Kusner et al., 2017; Dai et al., 2018; Winter et al., 2019; Griffiths & Hernández-Lobato, 2020; Dollar et al., 2021; Oliveira et al., 2022), generative adversarial networks (GANs) (Guimaraes et al., 2017; Sanchez-Lengeling et al., 2017), flow models (Shi et al., 2020) for one-shot generation, recurrent neural networks (RNNs) (Olivecrona et al., 2017; Bjerrum & Threlfall, 2017; Segler et al., 2018; Flam-Shepherd et al., 2021), and reinforcement learning approaches (You et al., 2018; Jin et al., 2020) for step-by-step generation. Some works (Li et al., 2019; Lim et al., 2020; Gómez-Bombarelli et al., 2018) aimed to preserve structural features such as molecular scaffolds or physicochemical properties like quantitative estimate of drug-likeness to improve the quality of generated molecules compared to random generation. Recently, several studies (Nesterov et al., 2020; Simm et al., 2020; Hoogeboom et al., 2022; Wu et al., 2022) have investigated ligand-based 3D molecular generation using a variety of techniques, including

VAEs, reinforcement learning, and diffusion-like methods, in conjunction with SE(3) equivariant models. However, these methods did not directly target the binding position and affinity against a specific protein pocket, resulting in generated molecules that may not perform well in real-world tasks.

**Pocket-Based Molecular Generation**   Binding affinity is a critical factor in drug design, leading to recent research that incorporates protein pocket information into molecular generation. Early efforts (Skalic et al., 2019; Xu et al., 2021) encoded pocket information as a condition for generating molecules in SMILES strings or molecular graphs. However, since binding affinity depends on the spatial positions of both the pocket and the molecule, subsequent studies focused on creating molecules with 3D structures.

Some approaches (Ragoza et al., 2022), known as 3D molecular density grid generation, converted pockets and molecules into 3D density grids and employed 3D convolutional models similar to image processing techniques. However, due to the large pocket cavity, the positions of pockets and molecules in 3D density grids are coarse-grained, resulting in information loss and difficulties in generating fine-grained molecules. Furthermore, this method is not end-to-end, as converting 3D density to 3D coordinates is necessary, which often leads to additional accuracy loss.

Other works (Luo et al., 2022; Liu et al., 2022; Peng et al., 2022), known as auto-regressive 3D molecular generation, sequentially sampled atoms in 3D space to form a molecule. However, this approach is inefficient due to the vast continuous 3D positional space. Moreover, unlike the sequential nature of text, atoms in a molecule lack a sequential order, making auto-regressive generation for 3D molecules less reasonable.

Recently, shape-based generation methods, such as (Adams & Coley, 2022; Long et al., 2022), define shapes in 3D space and train models to generate 3D molecules that fit into these shapes. Given a pocket, they first detect the shape of the pocket cavity and generate 3D molecules that can fit into that shape. However, besides shapes, the interactions between the atoms of the pocket and the molecule are also critical to binding affinity.

Lastly, some concurrent works have leveraged diffusion models, such as (Schneuing et al., 2022; Guan et al., 2023), to generate all molecular atoms in 3D space simultaneously. These works are similar to the initial version of `VD-Gen`, which directly moved atoms iteratively without involving virtual particles. However, these studies utilize diffusion mechanisms to alleviate training difficulties, whereas our method leverages optimal transport principles, specifically minimizing atom movement distances. Specifically, diffusion models usually employ a small-step denoising task, which results in stable and straightforward training due to the minimal distribution change at each step. For example, TargetDiff (Guan et al., 2023) directly leverages this small-step denoising method and can outperform previous models. However, TargetDiff also requires a significantly larger number of steps (1000 steps) to achieve satisfactory results. `VD-Gen` introduces more points (via a many-to-1 mapping) to alleviate the difficulties associated with the optimal transport in 1-to-1 mapping. This many-to-1 mapping allows for a more local optimization of the transport, considering only individual positions rather than global matching. Consequently, the training process becomes more stable. However, VD-Gen requires more points, resulting in a more complex pipeline that must merge the additional points. Experimental results, as detailed in Sec. C.3, demonstrate the superiority of our method in comparison to the diffusion baseline TargetDiff.

## 5   Conclusion

In this paper, we present `VD-Gen`, a novel pocket-based 3D molecular generation pipeline designed to generate fine-grained 3D molecules with high binding affinities against protein pockets in an end-to-end manner. Specifically, numerous virtual particles are initially distributed randomly within the pocket cavity, and subsequently moved iteratively to approximate the distribution of molecular atoms based on the training data. A 3D molecule is then extracted from these virtual particles using deep learning models. Following this, the atoms within the extracted molecule are further refined through iterative movement, resulting in a high-quality 3D molecule with fine-grained coordinates. Finally, a confidence score is predicted for the generated molecule for the need of selection and ranking. Experimental results demonstrate that `VD-Gen` outperforms other baseline methods in generating molecules with higher binding affinities to protein pockets and more accurate 3D binding structures. Additionally, we provide an ablation study, case study, and visualizations to further illustrate the effectiveness of `VD-Gen`.

**Reproducibility** We utilized CrossDocked data for the training, consistent with earlier models (Luo et al., 2022; Peng et al., 2022). Detailed formulations of our model can be found in Appendix A.1, while the training settings are elaborated in Appendix C.1.

**Broader Impact Statement** Our method aims to generate molecules with high binding affinity in drug discovery. However, it is critical to acknowledge and address the broader implications of this research from various perspectives, including drug safety and efficacy, dual-use concerns, and equitable development.

In terms of drug safety and efficacy, this research serves as a valuable starting point, but it does not guarantee the safety and efficacy of the resulting drug candidates. Further extensive experimental testing, including in vitro, in vivo, and clinical trials, will be required to confirm and validate the potential therapeutic benefits of the drug candidates.

Regarding dual-use concerns, we are aware that improved generative modeling of chemicals can carry risks of misuse, possibly leading to the development of harmful substances or drugs. As responsible researchers, we are committed to promoting and upholding ethical applications of our work.

Lastly, equitable development is a crucial aspect to consider when harnessing computational advances in drug development. We advocate for collaborative efforts and partnerships between researchers, industry, and policymakers to promote equitable access to the benefits of computational drug discovery.

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

# A  VD-Gen Details

Table 2: Symbols used in this paper.

| Symbol | Meaning |
|---|---|
| $\mathbf{P}$ | the set of atoms in the pocket |
| $\mathbf{V}_r$ | the set of virtual particles (VPs) that are generated at the $r$-th iteration in *Particle Movement* |
| $\mathbf{W}_r$ | the set of atoms that are generated at the $r$-th iteration in *Molecule Refinement* |
| $\mathbf{C}_p$ | the gridded 3D cubic of pocket atoms |
| $\mathbf{C}_m$ | the predicted cubic gridded cubic |
| $\mathbf{C}_m^g$ | the ground truth label for the gridded cubic |
| $\mathbf{G}$ | the set of ground-truth atoms |
| $\boldsymbol{x}_i^p$ | the $i$-th pocket atom's type (one-hot) |
| $\boldsymbol{y}_i^p$ | the $i$-th pocket atom's coordinate |
| $\boldsymbol{x}_i^g$ | the $i$-th ground-truth atom's type (one-hot) |
| $\boldsymbol{y}_i^g$ | the $i$-th ground-truth atom's coordinate |
| $\boldsymbol{x}_i^r$ | the $i$-th VP's type (one-hot) at the $r$-th iteration in *Particle Movement* |
| $\boldsymbol{y}_i^r$ | the $i$-th VP's coordinate at the $r$-th iteration in *Particle Movement* |
| $\bar{\boldsymbol{x}}_i^r$ | predicted atom type distribution of $i$-th VP at the $r$-th iteration |
| $a_i$ | The index of assigned target atom for the $i$-th VP in *Particle Movement* |
| $b_i$ | The index of assigned target atom for the $i$-th atom in *Molecule Refinement* |
| $\hat{\boldsymbol{x}}_i^r$ | the $i$-th atom's type (one-hot) at the $r$-th iteration in *Molecule Refinement* |
| $\hat{\boldsymbol{y}}_i^r$ | the $i$-th atom's coordinate at the $r$-th iteration in in *Molecule Refinement* |
| $\boldsymbol{q}$ | the pair representation of VP pair |
| $\boldsymbol{q}^l$ | the pair representation of VP pair at $l$-th layer |
| $\boldsymbol{q}^P$ | the pair representation of pocket atom pair |
| $\boldsymbol{q}^C$ | the pair representation of VP and pocket pair |
| $\bar{\boldsymbol{s}}$ | the predicted distance between VP and its target which is used to filter VP. |
| $\bar{\boldsymbol{r}}$ | predicted probability of merging type of VP pair |
| $\bar{m}$ | the predicted atom number |
| $k_{vp}$ | times of the number of atom, uses in *Particle Initialization* |
| $\boldsymbol{w}_i$ | The indices of VPs in the $i$-th cluster in *Molecule Extraction* |
| $\boldsymbol{h}$ | the node representation of VP |
| $\boldsymbol{h}^l$ | the node representation of VP at $l$-th layer |
| $\boldsymbol{h}^V$ | the node representation of VP in *Particle Movement* |
| $\boldsymbol{q}^V$ | the pair representation of VP pair in *Particle Movement* |
| $\boldsymbol{h}^W$ | the node representation of atom in *Molecule Refinement* |
| $\boldsymbol{q}^W$ | the pair representation of atom pair in *Molecule Refinement* |
| $\boldsymbol{h}^P$ | the node representation of pocket atom |
| $h_{an}$ | the model to predict atom number |
| $\boldsymbol{\theta}_{an}$ | the model parameter to predict atom number |
| $h_{pi}$ | the model to predict pocket cavity in *Particle Initialization* |
| $\boldsymbol{\theta}_{pi}$ | the model to predict pocket cavity in *Particle Initialization* |
| $\boldsymbol{\theta}_{pm}$ | the model parameter in *Particle Movement* |
| $\boldsymbol{\theta}_{me}$ | the model parameter in *Molecule Extraction* |
| $\boldsymbol{\theta}_{mr}$ | the model parameter in *Molecule Refinement* |
| $\boldsymbol{\theta}_{cp}$ | the model parameter in *Confidence Prediction* |
| $f_{pm}$ | the SE(3) backbone model, return types and coordinates of VPs in *Particle Movement* |
| $f_{me}$ | the SE(3) backbone model, return types and coordinates of atoms in *Molecule Refinement* |
| $L$ | the number of layers |

## A.1  Details of the SE(3) Backbone Model

Figure 5 illustrates the structure of the SE(3) backbone model used in `VD-Gen`. The abbreviations "Repr.", "Attn.", and "Dist." stand for "Representation", "Attention", and "Distance", respectively. The pocket encoder is displayed on the left side, which initially utilizes an atom-type embedding to encode the pocket atom type and a Gaussian kernel for encoding the pairwise distances between pocket atom pairs. Each layer of the pocket encoder incorporates a self-attention layer. On the right, the VP encoder is shown, which similarly

uses an atom-type embedding and a Gaussian kernel to encode the particle type and the pairwise distances between VPs. To facilitate interaction with the pocket encoder, an additional Gaussian kernel is used to encode the pairwise distances between VPs and pocket atoms. Within each layer of the VP encoder, a particle-pocket attention layer is incorporated prior to the self-attention layer, enabling interaction with the pocket encoder.

The subsequent paragraphs detail the components within the backbone model, as well as the overall pipeline of the backbone model, as presented in Algorithm 2. For the sake of simplicity, layer normalization has been omitted from the equations and algorithms.

**Gaussian Kernel** The pair-type-aware Gaussian kernel (Shuaibi et al., 2021; Zhou et al., 2023) is represented as:

$$\boldsymbol{p}_{ij} = \{\mathcal{G}(\mathcal{A}(d_{ij}, t_{ij}; \boldsymbol{a}, \boldsymbol{b}), \mu^k, \sigma^k) | k \in [1, D]\}, \mathcal{A}(d, r; \boldsymbol{a}, \boldsymbol{b}) = a_r d + b_r, \tag{5}$$

where $\mathcal{G}(d, \mu, \sigma) = \frac{1}{\sigma\sqrt{2\pi}} e^{-\frac{(d-\mu)^2}{2\sigma^2}}$ is a Gaussian density function with parameters $\mu$ and $\sigma$. $d_{ij}$ denotes the Euclidean distance of atom pair $ij$, and $t_{ij}$ refers to the pair type of atom pair $ij$. $\mathcal{A}(d_{ij}, t_{ij}; \boldsymbol{a}, \boldsymbol{b})$ represents the affine transformation with parameters $\boldsymbol{a}$ and $\boldsymbol{b}$, which transforms $d_{ij}$ according to its pair type $t_{ij}$.

**Pair Representation** Pair representation (Zhou et al., 2023) is used to further enhance the 3D spatial encoding. The pair representation is updated using the multi-head Query-Key product results in self-attention.

$$\boldsymbol{q}_{ij}^{l+1} = \boldsymbol{q}_{ij}^l + \{\frac{\boldsymbol{h}_i^l \boldsymbol{W}_{l,h}^Q (\boldsymbol{h}_j^l \boldsymbol{W}_{l,h}^K)^T}{\sqrt{d}} | h \in [1, H]\}, \tag{6}$$

where $\boldsymbol{h}_i^l$ denotes the atom/node representation of the $i$-th atom at the $l$-th layer, $\boldsymbol{q}_{ij}^l$ represents the pair representation of atom pair $ij$ in the $l$-th layer, $H$ is the number of attention heads, $d$ is the dimension of hidden representations, and $\boldsymbol{W}_{l,h}^Q$ ($\boldsymbol{W}_{l,h}^K$) is the projection for Query (Key) of the $l$-th layer $h$-th head.

To incorporate 3D information into the atom representation, pair representation is utilized in self-attention.

$$\boldsymbol{h}_i^{l+1,h} = \text{softmax}(\frac{\boldsymbol{h}_i^l \boldsymbol{W}_{l,h}^Q (\boldsymbol{h}_j^l \boldsymbol{W}_{l,h}^K)^T}{\sqrt{d}} + \boldsymbol{q}_{ij}^{l,h}) \boldsymbol{h}_j^l \boldsymbol{W}_{l,h}^V,$$
$$\boldsymbol{h}_i^{l+1} = \text{concat}_h(\boldsymbol{h}_i^{l+1,h}), \tag{7}$$

where $\boldsymbol{W}_{l,h}^V$ is the projection of Value for the $l$-th layer $h$-th head.

**Particle-Pocket Attention** The Particle-Pocket Attention is defined as follows:

$$\boldsymbol{h}_i^{l+1,h} = \text{softmax}\left(\frac{\boldsymbol{h}_i^l \boldsymbol{W}_{l,h}^{P,Q} (\boldsymbol{h}_j^P \boldsymbol{W}_{l,h}^{P,K})^T}{\sqrt{d}} + \boldsymbol{q}_{ij}^{C,l,h}\right) \boldsymbol{h}_j^P \boldsymbol{W}_{l,h}^{P,V},$$
$$\boldsymbol{h}_i^{l+1} = \text{concat}_h(\boldsymbol{h}_i^{l+1,h}),$$
$$\boldsymbol{h}_i^{l+1} = \boldsymbol{h}_i^l + g_1 \cdot \boldsymbol{h}_i^{l+1} + g_2 \cdot \text{MLP}(\boldsymbol{h}_i^{l+1}), \tag{8}$$

where $g_1$ and $g_2$ are learned parameters with initialized values of 0, $\boldsymbol{h}_j^P$ denotes the representation of the $j$-th pocket atom, $\boldsymbol{q}_{ij}^{C,l,h}$ represents the pair representation of particle-pocket pair $ij$ in the $l$-th layer $h$-th head, MLP refers to a fully-connected network with one hidden layer, and $\boldsymbol{W}_{l,h}^{P,Q}$, $\boldsymbol{W}_{l,h}^{P,K}$, and $\boldsymbol{W}_{l,h}^{P,V}$ are learnable projections for Query, Key, and Value, respectively.

**SE(3)-Equivariant Coordinate Head** Following (Zhou et al., 2023), the head can be denoted as:

$$\boldsymbol{y}_i^{r+1} = \boldsymbol{y}_i^r + \sum_{j=1}^n \frac{(\boldsymbol{y}_i^r - \boldsymbol{y}_j^r) z_{ij}}{n}, z_{ij} = \text{ReLU}((\boldsymbol{q}_{ij}^L - \boldsymbol{q}_{ij}^0) \boldsymbol{U}_1) \boldsymbol{U}_2, \tag{9}$$

---

**Algorithm 2** Backbone_Update

---

**Require: P**: pocket atoms, $\mathbf{V}_r$: virtual particles
1: $\boldsymbol{h}^{P,0} \leftarrow \text{Atom\_Type\_Embedding}(\mathbf{P})$         ▷ *Embeddings from atom types*
2: $\boldsymbol{q}^{P,0} \leftarrow \text{Gaussian\_Kernel}(\text{Dist\_Matrix}(\mathbf{P},\mathbf{P}))$     ▷ *Get invariant spatial positional embedding*
3:    *# Update Pocket Encoder*
4: **for** $l \in [1, ..., L)$ **do**
5:     $\boldsymbol{h}^{P,l}, \boldsymbol{q}^{P,l} \leftarrow \text{Self\_Attn}(\boldsymbol{h}^{P,l-1}, \boldsymbol{q}^{P,l-1})$     ▷ *Update by self attention*
6:     $\boldsymbol{h}^{P,l} \leftarrow \text{MLP}(\boldsymbol{h}^{P,l})$     ▷ *Update by Feed-Forward-Network*
7: $\boldsymbol{h}^P \leftarrow \boldsymbol{h}^{P,L}$
8: $\boldsymbol{h}^0 \leftarrow \text{Atom\_Type\_Embedding}(\mathbf{V}_r)$     ▷ *Embeddings from atom types*
9: $\boldsymbol{q}^0 \leftarrow \text{Gaussian\_Kernel}(\text{Dist\_Matrix}(\mathbf{V}_r,\mathbf{V}_r))$     ▷ *Get invariant spatial positional embedding*
10: $\boldsymbol{q}^{C,0} \leftarrow \text{Gaussian\_Kernel}(\text{Dist\_Matrix}(\mathbf{V}_r,\mathbf{P}))$     ▷ *Get invariant spatial positional embedding of particle-pocket pairs*
11:    *# Update Particle Encoder*
12: **for** $l \in [1, ..., L)$ **do**
13:     $\boldsymbol{h}^l, \boldsymbol{q}^l \leftarrow \text{Self\_Attn}(\boldsymbol{h}^{l-1}, \boldsymbol{q}^{l-1})$     ▷ *Update by self attention*
14:     $\boldsymbol{h}^l \leftarrow \text{MLP}(\boldsymbol{h}^l)$     ▷ *Update by Feed-Forward-Network*
15:     *# Only enabled at every 4-layer*
16:     **if** $l \mod 4 == 0$ **then**
17:       $\boldsymbol{h}^l, \boldsymbol{q}^{C,l} \leftarrow \text{Particle\_Pocket\_Attn}(\boldsymbol{h}^l, \boldsymbol{h}^P, \boldsymbol{q}^{C,l-1})$     ▷ *Update by Particle-Pocket Attention*
18: $\bar{\boldsymbol{x}}^{r+1} \leftarrow \text{Atom\_Type\_Head}(\boldsymbol{h}^L)$     ▷ *Atom Type Prediction*
19: $\boldsymbol{x}^{r+1} \leftarrow \text{sample}(\bar{\boldsymbol{x}}^{r+1})$     ▷ *Sample an atom type based on predicted probability*
20: $\boldsymbol{y}^{r+1} \leftarrow \text{SE(3)\_Head}(\boldsymbol{y}^r, \boldsymbol{q}^L)$     ▷ *Coordinate update*
21: **return** $\mathbf{V}_{r+1} = \{\boldsymbol{x}^{r+1}, \boldsymbol{y}^{r+1}\}, \boldsymbol{h}^L, \boldsymbol{q}^L$

---

where $n$ denotes the total number of atoms, $L$ represents the number of layers in the model, $\boldsymbol{y}_i^r \in \mathbb{R}^3$ is the input coordinate of the $i$-th atom, and $\boldsymbol{y}_i^{r+1} \in \mathbb{R}^3$ is the output coordinate of the $i$-th atom. The projection matrices $\boldsymbol{U}_1 \in \mathbb{R}^{H \times H}$ and $\boldsymbol{U}_2 \in \mathbb{R}^{H \times 1}$ are utilized to convert the pair representation to a scalar. It should be noted that the predicted coordinates are also used to calculate the distance between VPs as the predicted distance $\boldsymbol{d}_{ij}^r$ in Equation 1.

**Atom Type Prediction Head** A non-linear head with two layers is used to predict the atom type based on the atom representation in the last layer of the particle encoder:

$$\bar{\boldsymbol{x}}_i = \text{MLP}(\boldsymbol{h}_i^L) \tag{10}$$

where $\boldsymbol{h}_i^L$ represents the atom representation, and $L$ is the number of layers of the particle encoder.

### A.2 Pocket Cavity Detection by 3D U-Net

We use a 3D U-Net model Çiçek et al. (2016) to predict the cavity space within a protein pocket. To do this, we first define the pocket's bounding box (cube) and construct a 3D grid with cubic cells of length 2Å. Given the atoms of a pocket **P**, we create a 3D grid cubic with binary voxel values (where "1" indicates the presence of pocket atoms) to serve as the input for the 3D U-Net model. For the model's training target (the cavity space), we utilize the atoms in the ground-truth molecules, treating grids containing atoms as cavities.

Formally, we denote this process as $\mathbf{C}_m = h_{pi}(\mathbf{C}_p; \boldsymbol{\theta}_{pi})$, where $h_{pi}$ represents the 3D U-Net model with learnable parameter $\boldsymbol{\theta}_{pi}$, $\mathbf{C}_p \in \{0,1\}^{l_1 \times l_2 \times l_3}$ is the gridded 3D cubic (of size $l_1 \times l_2 \times l_3$) containing pocket atoms, and $\mathbf{C}_m \in \{0,1\}^{l_1 \times l_2 \times l_3}$ is the predicted cubic, with grids having a voxel value of 1 potentially representing cavities. The training objective function is grid-wise binary classification. Additionally, we employ a focal loss Lin et al. (2017) to alleviate the unbalanced classification problem:

$$\mathcal{L}_{pi} = \frac{1}{l_1 \times l_2 \times l_3} \sum_{i=1}^{l_1} \sum_{j=1}^{l_2} \sum_{k=1}^{l_3} \text{FL}(\bar{\boldsymbol{u}}_{i,j,k}, \boldsymbol{u}_{i,j,k}) \tag{11}$$

Here, FL denotes the focal loss function, $\boldsymbol{u}_{i,j,k}$ is the one-hot vector of voxel labels, and $\bar{\boldsymbol{u}}_{i,j,k}$ is the predicted vector obtained from model $h_{pi}$.

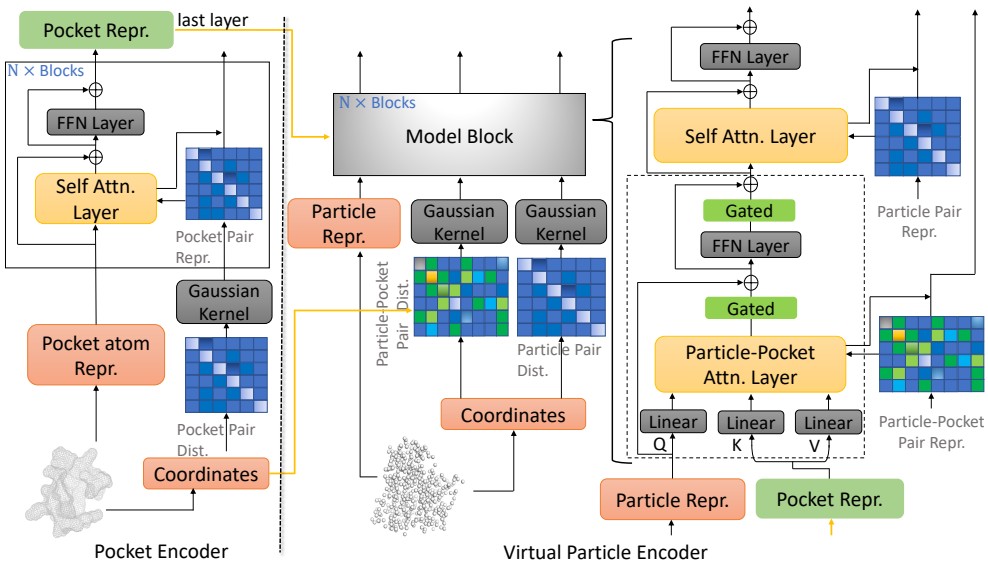

Figure 5: The backbone model used in `VD-Gen`. Details are in Appendix A.1 and Alg. 2.

### A.3    Training Loss in VD-Gen

**Binning Regression**    As described in Section 2.2, we convert several regression tasks into classification tasks by binning. The training loss for classification can then be written as:

$$\mathcal{L}_{binning\_regression} = \frac{1}{n} \sum_{i=1}^{n} \text{NLL}(\bar{z}_i, z_i) \tag{12}$$

where $\bar{z}_i$ represents the predicted probability distribution of bins, and $z_i$ denotes the one-hot vector of the target bin. During inference, the predicted value can be calculated from the predicted distribution over bins as follows:

$$\bar{z}_i = \sum_{k=1}^{n_{bin}} (bin\_val_k) \bar{z}_i[k], \tag{13}$$

where $n$ is the number of samples, $bin\_val_k$ represents the bin value of the $k$-th bin, $n_{\text{bin}}$ denotes the number of bins, $l$ is the size of each bin, and $\bar{z}_i[k]$ indicates the predicted probability of the $k$-th bin. Notably, the bin value is not the bin boundary value; instead, it is the average of the left and right boundaries.

**Focal Loss**    Several focal losses are also used in Section 2.2. Formally, these can be denoted as:

$$\mathcal{L}_{Focal} = \frac{1}{n} \sum_{i=1}^{n} \sum_{k=1}^{K} -z_i^k \log \bar{z}_i^k (1 - \bar{z}_i^k)^\gamma, \tag{14}$$

where $n$ is the number of samples, $K$ represents the number of types, $z_i$ denotes the one-hot vector of the target type, $\bar{z}_i$ indicates the predicted probability of the type, and $\gamma$ are hyper-parameters used to balance classes. In this paper, $\gamma$ is set to 2.

**LDDT**    The LDDT score is widely used in protein structure prediction (Mariani et al., 2013) and can be denoted as:

$$\text{LDDT} = \frac{1}{n} \sum_{i=1}^{n} \sum_{j \ eqi} \frac{1}{4} ((\text{err}_{ij} < 0.5) + (\text{err}_{ij} < 1.0)$$
$$+ (\text{err}_{ij} < 2.0) + (\text{err}_{ij}) < 4.0), \tag{15}$$

$$\text{where err}_{ij} = \text{L1}(||\hat{y}_i^{R_2} - \hat{y}_j^{R_2}||_2, ||\hat{y}_i^g - \hat{y}_j^g||_2), \tag{16}$$

$\hat{y}_i^{R_2}$ represents the predicted coordinate of the $i$-th particle after *Molecule Refinement*, and $\hat{y}_i^g$ denotes its ground truth coordinate.

---

**Algorithm 3** VD-Gen Inference Pipeline

---

**Require:** $R_1$, $R_2$: iterations in *Particle Movement* and *Molecule Refinement*, $k_{vp}$: times of the number of atom, **P**: pocket atoms with types and positions, $\mathbf{C}_p$: the gridded 3D cubic of pocket atoms, $h_{an}$: the model to predict atom number, $\boldsymbol{\theta}_{an}$: the model parameter to predict atom number, $h_{pi}$: the model to predict pocket cavity in *Particle Initialization*, $\boldsymbol{\theta}_{pi}$: the model parameter to predict pocket cavity in *Particle Initialization*, $\boldsymbol{\theta}_{pm}$, $\boldsymbol{\theta}_{me}$, $\boldsymbol{\theta}_{mr}$, $\boldsymbol{\theta}_{cp}$,: model parameters in *Particle Movement*, *Molecular Extraction*, *Molecular Refinement* and *Confidence Prediction*

1:
2:   *# Iterative Movement*
3:   **def** ITER_MOVE($R$, $\mathbf{V}_0$, $\mathbf{P}$, $\boldsymbol{\theta}$)):
4:       **for** $k \in [1,...,R]$ **do**
5:           $\mathbf{V}_k, \boldsymbol{h}^L, \boldsymbol{q}^L, \leftarrow$ Backbone_Update($\mathbf{V}_{k-1}, \mathbf{P}; \boldsymbol{\theta}$)           ▷ *backbone model update as in Alg. 2*
6:       **output** $\mathbf{V}_R, \boldsymbol{h}^L, \boldsymbol{q}^L$
7:   *# Particle Initialization*
8:   $\mathbf{C}_m = h_{pi}(\mathbf{C}_p, \boldsymbol{\theta}_{pi})$           ▷ *use 3D U-net model to predict the gridded cubic*
9:   $\bar{m} = h_{an}(\mathbf{P}, \boldsymbol{\theta}_{an})$           ▷ *Predict atom number*
10:  **for** $i$ in $[1,...,\bar{m}k_{vp}]$ **do**
11:      $\boldsymbol{x}_i^0 \leftarrow$ one_hot([MASK])           ▷ *The types of VPs are initialized as a meaningless [MASK] type*
12:      $\boldsymbol{y}_i^0 \sim$ Uniform($\mathbf{C}_m[\mathbf{C}_m == 1]$)   ▷ *the grids in the predicted gridded cubic with voxel value 1 are taken as cavity and the initial coordinates of VPs are uniformly sampled from the cavity space*
13:  $\mathbf{V}_0 = \{(\boldsymbol{x}_i^0, \boldsymbol{y}_i^0)\}_{i=1}^n$
14:
15:  *# Particle Movement*
16:  $\mathbf{V}_{R_1}, \boldsymbol{h}^V, \boldsymbol{q}^V \leftarrow$ Iter_Move($R_1, \mathbf{V}_0, \mathbf{P}, \boldsymbol{\theta}_{pm}$)           ▷ *Predict coordinates and types with iterative movement*
17:
18:  *# Molecule Extraction*
19:  $\bar{\boldsymbol{s}} \leftarrow$ Predict_Distance($\boldsymbol{h}^V; \boldsymbol{\theta}_{me}$)           ▷ *Predict the distance between VPs and the targets to filter particles*
20:  $\bar{\boldsymbol{r}} \leftarrow$ Predict_Merge($\boldsymbol{q}^V; \boldsymbol{\theta}_{me}$)           ▷ *Predict merging matrix*
21:  $\mathbf{W}_0 \leftarrow$ Molecule_Extraction($\mathbf{V}_{R_1}, \bar{m}, \bar{\boldsymbol{r}}, \bar{\boldsymbol{s}}$)
22:           ▷ *Filter and Merge VPs as in Alg. 4 using predicted merging matrix, atom number and predicted distance*
23:
24:  *# Molecule Refinement*
25:  $\mathbf{W}_{R_2}, \boldsymbol{h}^W, \boldsymbol{q}^W, \leftarrow$ Iter_Move($R_2, \mathbf{W}_0, \mathbf{P}; \boldsymbol{\theta}_{mr}$)           ▷ *Refine the coordinates and types*
26:  *# Confidence Prediction*
27:  Pred_LDDT $\leftarrow$ Predict_Confidence($\boldsymbol{h}^W; \boldsymbol{\theta}_{cp}$)           ▷ *Predict LDDT*
28:  **return** $\mathbf{W}_{R_2}$, Pred_LDDT           ▷ *Return the final positions and types, and the confidence score*

---

## A.4   VD-Gen Overall Algorithm

We summarize the overall inference pipeline of VD-Gen in Algorithm 3. First, a 3D U-Net model is used to predict the pocket cavity. Subsequently, the pocket encoder with a prediction head estimates the number of atoms based on the pocket representation. The main algorithm primarily relies on the "Iter_Move" function, which iteratively moves the VPs/atoms. Both *Particle Movement* and *Molecule Refinement* use the "Iter_Move" function for iterative refinement. In *Molecule Extraction*, two heads are used to predict the filtered probability and the pair merging probability, respectively, based on the node representation $\boldsymbol{h}^V$ and the pair representation $\boldsymbol{q}^V$. Based on these predictions, the "Molecule_Extraction" function constructs a molecule from VPs as described in Algorithm 4.

The training pipeline is very similar, except for the following differences. Firstly, to enhance efficiency, both $R_1$ and $R_2$ are sampled from the range of 1 to 4, and gradient backward is only activated during the final iteration. Secondly, in the *Molecule Extraction*, a teacher-forcing merging strategy (excluding binary search) is used for training. This entails utilizing the ground truth values for pair-wise merging probabilities and atom numbers, rather than generating predictions. Lastly, loss functions are enabled to obtain gradients essential for the training process.

---

**Algorithm 4** Molecule Extraction Algorithm

---

**Require:** $\mathbf{V}_{R_1} = \{(\boldsymbol{x}_i^{R_1}, \boldsymbol{y}_i^{R_1})\}_{i=1}^n$: virtual particles at $R_1$ iterations, $\bar{m}$: the predicted atom num, $\bar{\boldsymbol{r}} = \{(\bar{\boldsymbol{r}}_{ij})\}_{i=1,j=1}^{n \times n}$: the predicted merging probability matrix, $\bar{\boldsymbol{s}} = \{\bar{s}_i\}_{i=1}^n$: the predicted distance between VPs and target positions, $\zeta$: the filtering threshold

  1: *# Set the merge type between particles with the particle to be filtered to 0*
  2: **for** $i$ in $[1, ..., n]$ **do**
  3:     *# Filter the VPs according to predicted distance*
  4:     **if** $\bar{s}_i > \zeta$ **then**
  5:         $\bar{\boldsymbol{r}}[:, i] \leftarrow 0$                                 $\triangleright$ *Set $\{\bar{\boldsymbol{r}}_{i,j}\}_{j=1}^n$ to 0*
  6:         $\bar{\boldsymbol{r}}[i, :] \leftarrow 0$                                 $\triangleright$ *Set $\{\bar{\boldsymbol{r}}_{j,i}\}_{j=1}^n$ to 0*
  7: high $\leftarrow \max(\{(\bar{\boldsymbol{r}}_{ij})\}_{i=1,j=1}^{n \times n})$
  8: low $\leftarrow \min(\{(\bar{\boldsymbol{r}}_{ij})\}_{i=1,j=1}^{n \times n})$
  9: mid $\leftarrow \frac{\text{low} + \text{high}}{2}$
10: *# using the binary search to find the threshold*
11: **while** low $<$ high **do**
12:     $\boldsymbol{r}_{ij} \leftarrow \bar{\boldsymbol{r}}_{ij} > \text{mid}$
13:     $\mathbf{W}_0 \leftarrow [], \boldsymbol{w} \leftarrow [], m \leftarrow 0$
14:     *# Greedy merge based a random order*
15:     **for** $i$ in **random_perm**$(1, n)$ **do**
16:         $\boldsymbol{w}_i \leftarrow [],$
17:         **for** $j$ in $[1, ..., n]$ **do**
18:             **if** $\boldsymbol{r}_{ij} = \text{True}$ **then**
19:                 $\boldsymbol{r}[:, j] \leftarrow \text{False}$                $\triangleright$ *the merged particle will not be merged again*
20:                 $\boldsymbol{w}_i.\text{add}(j)$                    $\triangleright$ *add the particle indices into the i-th cluster*
21:         **if** len$(\boldsymbol{w}_i) > 0$ **then**
22:             $\hat{\boldsymbol{x}}_m^0 \sim \text{Uniform}(\{\boldsymbol{x}_k^{R_1} | k \in \boldsymbol{w}_i\})$             $\triangleright$ *Sample atom type from the merging list*
23:             $\hat{\boldsymbol{y}}_m^0 \leftarrow \text{Mean}(\{\boldsymbol{y}_k^{R_1} | k \in \boldsymbol{w}_i\})$   $\triangleright$ *the weighted average position according to the predicted distance is atom position after merging*
24:             $\mathbf{W}_0.\text{add}((\hat{\boldsymbol{x}}_m^0, \hat{\boldsymbol{y}}_m^0))$                      $\triangleright$ *add the atom to the set*
25:             $m \leftarrow m + 1$                           $\triangleright$ *Count the number of clusters*
26:     *# find the threshold*
27:     **if** $m = \bar{m}$ **then**
28:         break
29:     **else**
30:         **if** $m < \bar{m}$ **then**
31:             low $\leftarrow$ mid             $\triangleright$ *too many particles are merged, the threshold needs to be increased*
32:         **else**
33:             high $\leftarrow$ mid            $\triangleright$ *Too few particles are merged, the threshold needs to be lowered*
    **return** $\mathbf{W}_0$                                                  $\triangleright$ *Return particle set after merging*

---

### A.5 Molecule Extraction Algorithm

The process of merging VPs into atoms is presented in Algorithm 4, where a binary search technique is used to identify a merging threshold. To reduce training costs, teacher-forcing is utilized, thereby eliminating the need for binary search during training. Instead of predicting pair-wise merging probabilities and the number of atoms, their ground truth values are directly used. During inference, the binary search technique is applied. Furthermore, to account for potential errors in atom number prediction, a range ($\pm 10$) of atom numbers is explored, with the final selection based on confidence scores.

## B The Failed Attempt

In the initial version of VD-Gen, virtual particles were not incorporated into the pipeline. Specifically, there were no *Particle Movement* and *Molecule Extraction* stages, and the number of initial particles was equal to the predicted atom number $\bar{m}$. Furthermore, to ensure that each particle had a target atom, we employed

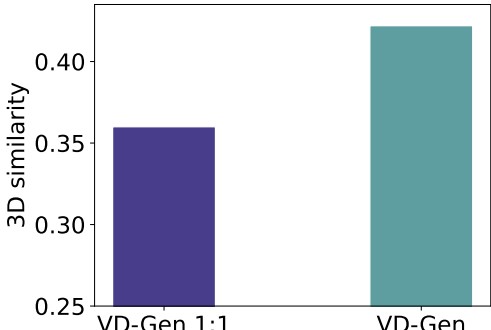

Figure 6: Comparison for different training frameworks. An initial attempt, called "VD-Gen 1:1", involves employing an equal number of particles and atoms, maintaining a 1:1 particle-atom bijective mapping, and iteratively moving particles without the VP-to-atom extraction process. Results demonstrate the effectiveness of the proposed `VD-Gen` framework.

the Wasserstein distance to obtain an optimal bijective mapping between particles and target atoms (solved by Hungarian algorithm). While this approach appeared more straightforward than the current `VD-Gen`, it proved to be less effective. More specifically, we observed unstable training (i.e., divergence) in this setting. A closer examination revealed that when randomly positioning the $\bar{m}$ virtual particles, some were close to the ground-truth positions, while others were far from their intended destinations. This discrepancy led to training instability: setting a long movement iteration caused some "near" particles to fail to learn to stay, while a short movement iteration prevented "far" particles from reaching their destinations. This issue is closely related to the high variance and slow convergence of the Wasserstein distance with respect to the number of samples, as discussed in (Dudley, 1969; Weed & Bach, 2019).

To address this problem, we introduced virtual particles(VPs), ensuring that there were more VPs than atoms. By employing more VPs and the nearest target assignment, each target atom was highly likely to have several VPs in close proximity. These VPs could then be moved to the target atom using fewer and similar movement iterations. Once the VPs were moved, they clustered around the positions of the target atoms. A clustering algorithm could then be utilized to extract atoms from the VPs. Although some VPs might fail, this would not impact the clustering results, leading to stable training.

We conducted an additional experiment to elucidate the superior performance attained by utilizing VPs. Figure 6 illustrates that the performance of the initial version of `VD-Gen` (referred to as "VD-Gen 1:1") is considerably inferior to that of the current version.

## C    Experiment Details and More Results

### C.1    Training Details

The detailed configurations of `VD-Gen` are listed in Table 3 and Table 4 [1]. While a multitude of hyperparameters are introduced in our approach, only a select few are of paramount importance. As depicted in Table 4, beyond standard hyperparameters—such as layer count, hidden dimensions, and learning rate—we introduce additional parameters: six loss coefficients, three stability thresholds denoted by $\tau$, $\delta$, and $\zeta$, parameters $R_1$ and $R_2$ representing iterations in Particle Movement and Molecule Refinement respectively, and $k_{vp}$ which indicates a multiplicative factor. Among these, $R_1$, $R_2$, and $k_{vp}$ are especially important and were carefully tuned as shown in our Ablation Study (Section 3.3). For the other hyperparameters, we chose values based on previous experience since their impact on model performance seems small. Notably, the model's training stability was maintained even without extensive tuning of these parameters. Our approach's effectiveness is further supported by empirical evidence: VD-Gen consistently outperforms previous benchmarks, showing its relative insensitivity to hyperparameter changes.

---

[1]The codes of the 3D U-net model are implemented based on https://github.com/wolny/pytorch-3dunet

Table 3: Settings for the 3D U-Net models in `VD-Gen`.

| Name | Value |
|---|---|
| Number of U-Net encoders | 5 |
| Number of U-Net decoders | 5 |
| Output channels in each encoder | 16, 32, 64, 128, 256 |
| Convolution kernel size | 3 |
| Pooling kernel size | 2 |
| Batch size | 16 |
| Max training steps | 500k |
| Warmup steps | 20K |
| Peak learning rate | 2e-4 |
| Adams $\epsilon$ | 1e-6 |
| Adams($\beta_1$, $\beta_2$) | (0.9,0.99) |
| Gradient clip norm | 0.5 |

## C.2 Evaluation Metrics

- *3D similarity.* We use LIGSIFT (Roy & Skolnick, 2015) for calculating 3D similarity. By default, LIGSIFT aligns the input molecules before computing 3D similarity. However, we aim to assess the generated 3D structure directly to examine the end-to-end performance. Consequently, we *remove the alignment feature in LIGSIFT.*

- *Vina.* We utilize AutoDock Vina1.2 (Eberhardt et al., 2021) to obtain Vina scores. Specifically, re-docking is applied, meaning that the binding pose and the conformation of the ligand molecule generated by the model are disregarded, and a new binding pose and molecular conformation are re-calculated by AutoDock Vina1.2. We argue that re-docking in Vina may not accurately reflect the actual performance of pocket-based 3D molecular generation. However, for consistency with previous work, we retain it as one of the metrics.

- *Vina\*.* Vina\* refers to Vina without re-docking. We use the built-in energy optimization process based on the Vina scoring function in AutoDock Vina1.2 (Eberhardt et al., 2021) to minimize the energy of the binding pose of generated molecules. Subsequently, we use the Vina scoring function to score the energy-minimized binding pose, resulting in the Vina\* score.

- *MM-PBSA.* We adopt the default settings of parameters (i.e., solvation mode: GB-2(Onufriev et al., 2004), protein forcefield: amber03(Duan et al., 2003), ligand charge method: bcc(Jakalian et al., 2000), dielectric constant: 4.0) and workflow (i.e., force field building, structure optimization by energy minimization, MM/GB(PB)SA calculation) of (Yang et al., 2022) to compute MM-PBSA scores. As the crystal structure indicates the preferred binding pose against a specific target, we filter the generated molecules based on their 3D similarity to the molecule in the crystal structure and consider molecules with a 3D similarity score above 0.4 as effective. We only calculate the MM-PBSA score for these effective molecules. In Table 6, we present MM-PBSA S.R. (success rate), which calculates the proportion of effective MM-PBSA scores of the generated molecules. For MM-PBSA B.T. and MM-PBSA Rank we have:

$$\text{MM-PBSA B.T.} = \frac{1}{n_p} \sum_{i=1}^{n_p} \frac{|\{g \in \mathcal{M}_i | \text{MM-PBSA}(g) < \text{MM-PBSA}(T_i)\}|}{|\mathcal{M}_i|},$$

$$\text{MM-PBSA Rank} = \frac{1}{n_p} \sum_{i=1}^{n_p} \text{rank}_i,$$

where $n_p$ denotes the number of proteins in the test set, $\mathcal{M}_i$ represents the generated molecular set of the $i$-th protein, $T_i$ denotes the ground-truth molecule in the crystal structure of the $i$-th protein, and $\text{rank}_i$ represents the ranking index of the current model among all compared models under the $i$-th protein, ranked by MM-PBSA.

- Metric for ablation studies. For ablation studies, we use the 3D similarity between the generated molecules and the ground truth as the metric, since it reflects the generative ability based on the pocket structure, and Table 1 shows a strong correlation between 3D similarity and binding affinity.

Table 4: Settings for SE(3) models in `VD-Gen`.

| Name | Value |
|---|---|
| Training | |
| Particle encoder layers | 12 |
| Pocket encoder layers | 15 |
| Particle-Pocket Attention layers | 3 |
| Peak learning rate | 5e-5 |
| Batch size | 32 |
| Max training steps | 100k |
| Warmup steps | 10K |
| Attention heads | 64 |
| FFN dropout | 0.1 |
| Attention dropout | 0.1 |
| Embedding dropout | 0.1 |
| Weight decay | 1e-4 |
| Embedding dim | 512 |
| FFN hidden dim | 2048 |
| Gaussian kernel channels | 128 |
| Activation function | GELU |
| Learning rate decay | Linear |
| Adams $\epsilon$ | 1e-6 |
| Adams($\beta_1$, $\beta_2$) | (0.9,0.99) |
| Gradient clip norm | 1.0 |
| Loss weight of $\mathcal{L}_{an}$ in *Particle Initialization* | 1.0 |
| Loss weight of *Particle Movement* | 1.0 |
| Loss weight of $\mathcal{L}_{Merge}$ in *Molecule Extraction* | 10 |
| Loss weight of $\mathcal{L}_{error\_pred}$ in *Molecule Extraction* | 0.01 |
| Loss weight of *Molecule Refinement* | 1.0 |
| Loss weight for *Confidence Prediction* | 0.01 |
| $\tau$, the clip value for coordinate loss | 2.0 |
| $\delta$, the threshold for coordinate regularization | 1.0 |
| $\zeta$, the filtering threshold in *Molecule Extraction* | 2.0 |
| $R_1$, Iterations in *Particle Movement* | sampled from [1, 4] |
| $R_2$, Iterations in *Molecule Refinement* | sampled from [1, 4] |
| $k_{vp}$ times of the number of atom, uses in *Particle Initialization* | sampled from [16.0, 18.0] |
| Inference | |
| $R_1$, Iterations in *Particle Movement* | 4 |
| $R_2$, Iterations in *Molecule Refinement* | 16 |

## C.3 More Results

In Table 5, we report more percentile results for Vina, Vina*. In Table 6, we report more percentile MM-PBSA results and MM-PBSA S.R. scores. The MM-PBSA S.R. scores in many baselines are very low. Thus, there are not enough effective MM-PBSA results to calculate percentile results in some baselines. Therefore, in each pocket, we replace the failed MM-PBSA result with the worst one generated by that baseline. And we calculated the percentile results after the replacement.

In Table 7, we compare `VD-Gen` with a diffusion-based model, TargetDiff Guan et al. (2023). From the result, we can find that our proposed `VD-Gen` is better than TargetDiff. We also present some cases in Fig. 7 to demonstrate the difference of generated molecules between `VD-Gen` and TargetDiff. It can be observed that, under similar Vina scores, the molecules generated by VD-Gen exhibit higher 3D similarity to the original molecules, better MM-PBSA results, and topological structures more closely related to the original molecules compared to those generated by TargetDiff. Furthermore, some unreasonable structures are present in the molecules generated by TargetDiff. For instance, the leftmost molecule forms a rare 15-membered ring structure in drug molecules, which is predominantly composed of aliphatic bonds, implying a high degree of

flexibility. Such a molecule is unlikely to stably bind with the target protein. Additionally, the presence of two consecutive carbon-carbon double bonds in the molecule represents an unreasonable chemical structure.

Table 5: More results on Vina and Vina*. We additionally computed the standard deviations for VD-Gen's results.

| Model | 5-th | | 10-th | | 25-th | | 50-th | |
|---|---|---|---|---|---|---|---|---|
| | Vina($\downarrow$) | Vina*($\downarrow$) | Vina($\downarrow$) | Vina*($\downarrow$) | Vina($\downarrow$) | Vina*($\downarrow$) | Vina($\downarrow$) | Vina*($\downarrow$) |
| LiGAN (Ragoza et al., 2022) | -6.724 | -5.372 | -6.324 | -4.922 | -5.740 | -4.215 | -5.065 | -3.49 |
| 3DSBDD (Luo et al., 2022) | -8.662 | -7.227 | -8.296 | -6.664 | -7.557 | -5.633 | -6.474 | -4.078 |
| GraphBP (Liu et al., 2022) | -8.710 | -3.689 | -7.832 | -2.774 | -6.765 | -1.169 | -5.625 | -1.2 |
| Pocket2Mol (Peng et al., 2022) | -8.332 | -6.525 | -8.015 | -5.399 | -7.467 | -3.513 | -6.837 | -1.808 |
| VD-Gen | **-8.998** | **-7.404** | **-8.570** | **-6.737** | **-7.891** | **-5.732** | **-7.207** | **-4.549** |
| | $\pm$0.0009 | $\pm$0.0257 | $\pm$0.0024 | $\pm$0.0013 | $\pm$0.0017 | $\pm$0.016 | $\pm$0.0038 | $\pm$0.015 |

Table 6: More MM-PBSA results. We additionally computed the standard deviations for VD-Gen's results.

| Model | 5-th MM-PBSA($\downarrow$) | 10-th MM-PBSA($\downarrow$) | 25-th MM-PBSA($\downarrow$) | 50-th MM-PBSA ($\downarrow$) | MM-PBSA-S.R.(%$\uparrow$) |
|---|---|---|---|---|---|
| LiGAN (Ragoza et al., 2022) | -18.462 | -13.374 | -8.775 | -7.418 | 11.9 |
| 3DSBDD (Luo et al., 2022) | -30.560 | -23.623 | -13.544 | -7.739 | 12.9 |
| GraphBP (Liu et al., 2022) | -5.579 | -4.894 | -4.894 | -4.894 | 0.2 |
| Pocket2Mol (Peng et al., 2022) | -8.226 | -5.945 | -5.398 | -5.398 | 1.8 |
| VD-Gen | **-50.744** | **-44.268** | **-35.178** | **-25.191** | **43.106** |
| | $\pm$0.138 | $\pm$0.186 | $\pm$0.168 | $\pm$0.137 | $\pm$ 0.015 |

Table 7: Comparison with diffusion-based models. We additionally computed the standard deviations for VD-Gen's results.

| Model | 3D Sim($\uparrow$) | Vina($\downarrow$) | Vina*($\downarrow$) | MM-PBSA($\downarrow$) | MM-PBSA-B.T.(%$\uparrow$) |
|---|---|---|---|---|---|
| TargetDiff (Guan et al., 2023) | 0.368 | -8.710 | -7.025 | -35.46 | 4.3 |
| VD-Gen | **0.422** | **-8.998** | **-7.404** | **-50.744** | **11.623** |
| | $\pm$0.0001 | $\pm$0.0009 | $\pm$0.0257 | $\pm$0.138 | $\pm$0.163 |

## C.4 Diversity Results

In Tabel 8, we report the diversity of molecules generated by VD-Gen and other baselines. Specifically, we report the proportions of unique SMILES and scaffolds present in the generated molecules. The findings reveal that the molecules produced by VD-Gen exhibit greater diversity compared to those generated by other baselines, particularly in terms of scaffold diversity.

Table 8: More results on Diversity. We additionally computed the standard deviations for VD-Gen's results.

| Model | LiGAN | 3DSBDD | GraphBP | Pocket2Mol | TargetDiff | VD-Gen |
|---|---|---|---|---|---|---|
| SMILES Diversity (↑) | 92.3 | 83.98 | 54.17 | 61.44 | **96.2** | 95.36 ± 0.033 |
| Scaffold Diversity (↑) | 54.87 | 55.95 | 50.82 | 36.35 | 61.6 | **79.46** ± 0.169 |

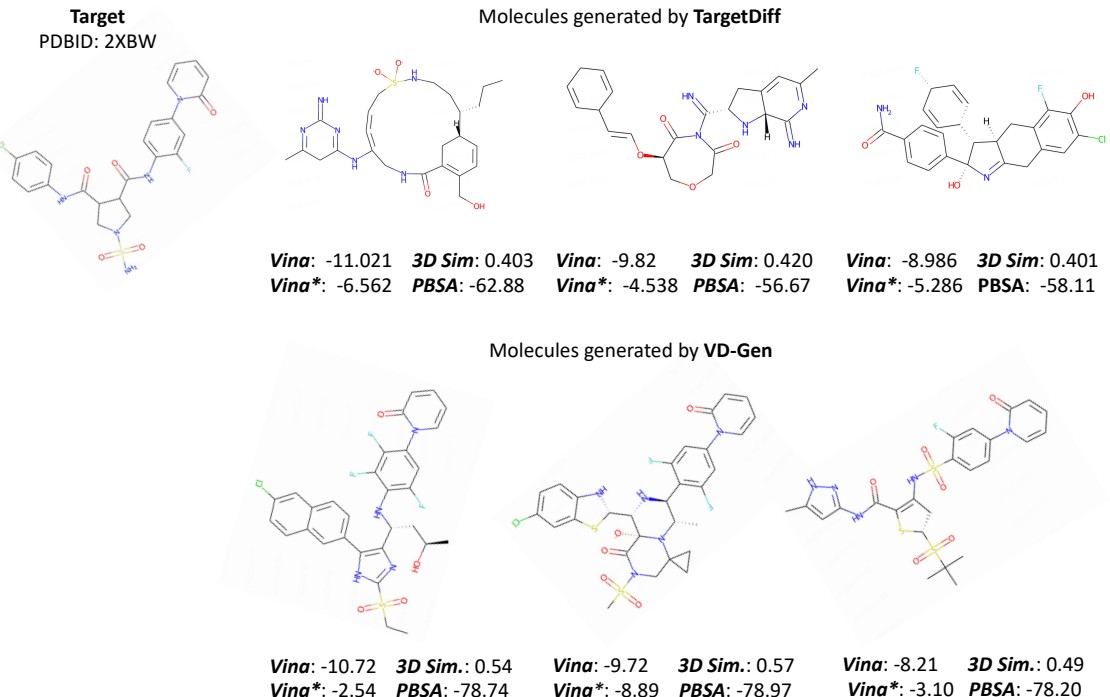

Figure 7: More examples for the generated molecules. We show the molecular graphs of the generated molecules of VD-Gen and TargetDiff with high 3D similarity to the reference molecule for the target (PDBID 2XBW) along with Vina score and MM-PBSA.

Table 9: Inference Efficiency.

| Model | 3DSBDD | GraphBP | Pocket2Mol | TargetDiff | VD-Gen |
|---|---|---|---|---|---|
| Time(s)($\downarrow$) | 14.153 | 1.660 | 3.476 | 30 | 3.678 |

## C.5 Inference Efficiency

The experimental results demonstrate the effectiveness of the proposed VD-Gen, and we also evaluate its efficiency. Specifically, we compare the inference speed of generating a single molecule for 3DSDBB, GraphBP, Pocket2Mol, and our VD-Gen. The results are summarized in Table 9. TargetDiff (Guan et al., 2023) exhibits the slowest performance, primarily due to its diffusion sampling times being set at 1000. 3DSDBB is the slowest AR method due to its inefficient MCMC sampling. Although GraphBP is the fastest, it generates the lowest-quality molecules. VD-Gen and Pocket2Mol exhibit similar efficiency levels, but VD-Gen significantly outperforms Pocket2Mol in terms of effectiveness. Given the large number of VPs and multiple movement iterations, it is expected that VD-Gen will not be the fastest method. We leave the efficiency improvement to future work.

## C.6 Molecular Optimization Results

**Differences in Training Molecular Optimization Models**   Training the molecular optimization model involves several modifications. The specific pipeline for this task is illustrated in Figure 8. Key changes include:

- Instead of predicting the entire molecule, in this task, our model aims to predict only a portion of it. To achieve this, 25% to 40% of atoms are removed from the original molecule, with the remaining atoms serving as input for prediction.

- During training, the number of VPs is significantly reduced, amounting to only 8 times the number of actual atoms.

- VPs are not uniformly distributed within the pocket cavity; instead, they are distributed around the removed atoms.

**Experiment**   We compare our model with a traditional molecular fragments optimization model Deep-Frag (Green & Durrant, 2021). DeepFrag can replace molecular fragments based on SMILES, which is a 1D model without pocket information. The results are shown in Table 10 and Table 11. From them, it is clear that VD-Gen can outperform the baseline in molecular optimization.

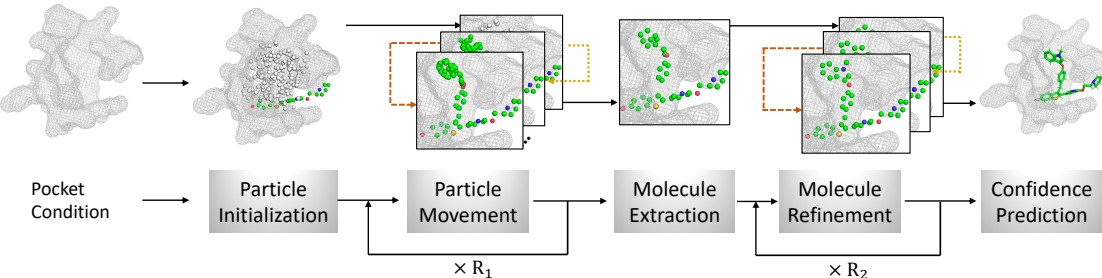

Figure 8: Extending VD-Gen to molecular optimization.

Table 10: Full percentile results on Vina and Vina*, in molecular optimization tasks. We additionally computed the standard deviations for VD-Gen's results.

| Model | 5-th | | 10-th | | 25-th | | 50-th | |
|---|---|---|---|---|---|---|---|---|
| | Vina($\downarrow$) | Vina*($\downarrow$) | Vina($\downarrow$) | Vina*($\downarrow$) | Vina($\downarrow$) | Vina*($\downarrow$) | Vina($\downarrow$) | Vina*($\downarrow$) |
| DeepFrag | -8.357 | - | -8.132 | - | -7.775 | - | -7.372 | - |
| VD-Gen | **-8.870** | -7.484 | **-8.592** | -7.095 | **-8.149** | -6.339 | **-7.677** | -5.334 |
| | ±0.002 | ±0.0119 | ±0.0022 | ±0.0009 | ±0.0019 | ±0.0049 | ±0.0033 | ±0.0067 |

Table 11: Full percentile results on MM-PBSA, in molecular optimization tasks. We additionally computed the standard deviations for VD-Gen's results.

| Model | 5-th | 10-th | 25-th | 50-th | |
|---|---|---|---|---|---|
| | MM-PBSA($\downarrow$) | MM-PBSA($\downarrow$) | MM-PBSA($\downarrow$) | MM-PBSA ($\downarrow$) | MM-PBSA B.T.($\uparrow$) |
| DeepFrag | -51.783 | -48.959 | -39.786 | -34.485 | 23.9 |
| VD-Gen | **-53.270** | **-50.387** | **-45.403** | **-37.226** | **29.987** |
| | ±0.123 | ±0.255 | ±0.241 | ±0.169 | ±0.181 |

