# OpenReview forum: "3D Molecular Generation via Virtual Dynamics"
_TMLR — Accepted by TMLR_

### Review · Reviewer_XPHM · 2023-08-29

**Summary Of Contributions:**

This work proposes an algorithm called VD-gen to generate small molecules in protein binding pockets. The model has several stages:

- First, a neural network predicts the location of the binding site and initializes the virtual particles
- Second, another neural network evolves the locations of the virtual particles by using a neural network to update their positions for several time steps
- Third, a post-processing step is done to convert the virtual particles into atoms using a hierarchical clustering procedure

The authors evaluate their method on a test dataset from a past paper (Yang et al 2022), showing favourable performance.

**Audience:**

Yes

**Broader Impact Concerns:**

no concerns

**Claims And Evidence:**

Yes

**Requested Changes:**

In general I think the paper is good, so most of these changes are suggestions rather than requirements for me to recommend acceptance.

**Important**

To me the most important change is discussing / relating the work to diffusion models more because of the apparent similarity with VD-gen. I think it would be useful to both discuss it qualitatively, but also add representative SOTA diffusion models as baselines to Table 1 (although if the authors really don't want to run extra baselines then I won't insist on it).

**Suggested changes**

- Clarify function definitions more. I found the input/output spaces of some functions (e.g. $M$ in section 2) to be unclear. I suggest the authors use the $\mapsto$ notation to define functions: e.g. $M:\mathcal{X}\mapsto\mathcal{Y}$.
- Double check citations: I saw Feynman (2017) cited, but Feynman died in 1988
- Add error bars or standard deviations to Table 1: since these methods have randomness it is important to characterize the scale of the randomness vs the scale of the difference between the performance of different methods. If the scales are comparable then I think a statistical test is needed to make claims about performance differences.
- Could explain "Wasserstein distance" approach in paragraph 2 of section 2 a bit more. I think I got it but it might not be clear to other readers. E.g. "One approach to train M would be to use supervised learning to minimize the error between the produced atoms and the ground truth atoms. However, this requires assigning each produced atom to an at
om in the ground truth set. While this could be done by choosing an assignment with minimal Wasserstein distance, ..."
- Remove the phrase "due to space constraints". TMLR does not have space constraints.
- Does SE(3) model mean SE(3) _equivariant_ model? If so then maybe change "SE(3) model" to "SE(3) equivariant model"

If you do make changes, I think myself and the other reviewers would appreciate you highlighting these changes in the revised pdf, e.g. by using `{\color{red}}` to make the text red.

**Questions**

- In section 2.2, when assigning virtual particles, do you ensure that every ground truth atom has at least 1 assigned virtual particle? The description in the text made me think this is not the case
- Why do you bucket things and turn them into classification tasks instead of leaving as regression? (e.g. equations 2,4)
- Why does the magnitude of "MM-PBSA" vary so much between methods in Table 1?
- Case study in 3.4: were these proteins in the training set (or similar proteins)? How similar are these examples to examples in the training set? In general it is easy for models to regurgitate things from the training set.
- How are the confidence scores used by the algorithm? I could not find this in the text.
- Why is $M$ a deterministic function and not a distribution?

**Strengths And Weaknesses:**

Strengths:
- paper is well-written and well-organized. I appreciate that it is under 12 pages.
- targets and interesting and relevant real-world problem
- good experimental results
- includes interesting ablation studies and visualization. I think this gives good insight into the model's behaviour, and is definitely better than most standard ML papers that just report average error metrics

Weaknesses:
- In my opinion, the pipeline is very complex and has lots of hyperparameters. It seems like the behaviour of the algorithm would be very sensitive to this. In particular, the merging procedure (step 3) seems "hacky". This is a subjective weakness though.
- The work seems very similar to diffusion models in that it evolves a set of particles over time, but the relationship to diffusion models is only discussed very briefly. Diffusion models have the advantage of having a principled end-to-end training procedure and nice probabilistic interpretation. It would be good to get the author's perspective of what pros/cons VD-gen has with respect to diffusion models. This could also include more experimental comparison.
- Tables have no error bars

---

> ### Author Response · Authors · 2023-11-01
> **Response to Reviewer XPHM (1/2)**
>
> We sincerely thank the reviewer for their insightful comments and feedback on our paper. We have carefully addressed each concern as detailed below:
>
> 1. **Pipeline Complexity & Hyperparameters (Weakness 1):**
>
> While we acknowledge the complexity of our pipeline—especially in the molecular extraction stage, its training is not very sensitive. As outlined in Table 4, our model introduces several hyperparameters beyond the standard ones like layer count, hidden dimensions, and learning rate. Specifically, we introduce six loss coefficients, three stability thresholds labeled as $\tau$, $\delta$, and $\zeta$, along with parameters $R_1$ and $R_2$ that signify iterations in Particle Movement and Molecule Refinement, respectively. We also incorporate $k_{vp}$, a multiplicative factor.
>
> Of these, $R_1$, $R_2$, and $k_{vp}$ are especially critical. These parameters were tuned, as detailed in our Ablation Study. For the other hyperparameters, we relied on our prior experience to select values. It's worth noting that even in the absence of extensive tuning for these parameters, the model's training stability remained well. The empirical results further vindicate our method: VD-Gen consistently surpasses previous benchmarks.
>
>
> 2. **Comparison with Diffusion Models (Weakness 2):**
>
> In our initial submission, we included a comparison with TargetDiff in Table 7. We've now provided an in-depth discussion on diffusion models in the "Related Work" section.
>
> 3. **Error Bars in Table 1 (Weakness 3 & Requested Changes):**
>
> We appreciate the suggestion regarding error bars. We didn't include the error bar for the following reasons:
>
>   - VD-Gen, 3DSBDD, and Pocket2Mol inherently rank and select generated molecules, minimizing randomness. For instance, 3DSBDD and Pocket2Mol use beam-search for consistent results, while VD-Gen utilizes predicted confidence scores for top molecule selection.
>
>   - Each method selects 100 molecules from a 500-molecule pool per pocket, as detailed in Section 3.2. Table 1 reports the 5th percentile results, with other percentiles in Appendix C.3. Given these metrics reflect aggregate results, they sufficiently account for any randomness.
>
> We can add the error bass if the reviewer thinks it is necessary.
>
> 4. **Addressing Requested Changes:**
>   - Apologies for any confusion caused by our previous notations. Notations have been revised for clarity and consistency.
>   - We apologize for the citation problem. We have now updated our citation to reflect the original version, and have also double-checked all other citations.
>   - The details of "Wasserstein distance" approach have been included for clarity in Section 2.
>   - We appreciate your input on the "due to space constraint" phrase. We have removed this statement from our paper.
>   - You are correct that "SE(3) model" means "SE(3) equivariant model". We have updated the paper accordingly to reflect this.

---

> ### Author Response · Authors · 2023-11-01
> **Response to Reviewer XPHM (2/2)**
>
> 5. **Answers to Questions:**
>   - While we don't ensure every ground truth atom gets a virtual particle in each trial, the model undergoes multiple passes with varied initializations. This means over time, there's a strong probability that each atom receives virtual particles, addressing the highlighted concern and ensuring reliable outputs.
>   - We've adopted pLDDT from AlphaFold2 which transitions regression to classification, benefiting from enhanced training stability. Equations 2 and 4, predicting confidence scores, align with this classification model. We've revised our paper for clarity.
>   - As MM-PBSA scores can vary significantly between complexes, their magnitudes may also differ for various baselines. Consequently, we introduced the MM-PBSA Rank metric to enable a more effective comparison of models.
>   - To avoid data leakage, we indeed exclude complexes from the training data that share similar protein sequences with those in the test set, as mentioned in Section 3.1 (Data paragraph).
>   - Confidence scores are employed to rank or select generated molecules. In our experiment, as outlined in Section 3.2 (Baselines paragraph), we utilize pLDDT to select the top 100 molecules from the 500 randomly generated molecules.
>   - We apologize for any confusion caused by our previous notation. We believe the reviewer inquired about the function $f$, not $\mathbf{M}$. The reviewer is correct that $f$ generates a distribution rather than a deterministic output. We have revised the paper to clarify this point.

---

> > ### Comment · Reviewer_XPHM · 2023-11-06
> > **Left another comment with my response**
> >
> > Thank you for addressing the points I raised in my review. I left another comment [here](https://openreview.net/forum?id=QvipGVdE6L&noteId=MmeV1HqTfQ) with a more detailed response. I am happy with most of the changes, but had some remaining issues with the notation, discussion of diffusion models, and error bars.

---

### Review · Reviewer_Qyn7 · 2023-10-24

**Summary Of Contributions:**

This paper proposes a new 3D molecular generation pipeline based on binding pockets of proteins in an end-to-end manner. In particular, numerous virtual particles are randomly sampled within the pocket cavity first, which will be iteratively moved to approximate the distribution of molecular atoms. From these moved virtual particles, a 3D molecule will be extracted and further refined through another iterative process. Finally, candidates will be ranked according to LDDT scores and the one with the highest score will be selected. As far as I know, approximating the molecular atom distribution through the iterative movement of virtual particles has not been studied before. The experimental results show that the proposed model achieves the highest performance on all the metrics.

**Audience:**

Yes

**Claims And Evidence:**

Yes

**Requested Changes:**

1. The writting of this paper really needs to be refined, especially the design intuition of some important components.

2. The author may need to provide a figure illustrating the overall architecture of the proposed method.

3. The author mentioned in Introduction that their model can generate diverse drug-like molecules. However, I have't seen any diversity metric about the designed molecules. The author may need to provide additional diversity and novelty about the designed molecules.

**Strengths And Weaknesses:**

The strengths and weaknesses of this paper are listed as follows:

**strengths:**

1.  As far as I know, approximating the molecular atom distribution through the iterative movement of virtual particles has not been studied before. Therefore, the proposed method should be novel to some extent.

2. The proposed method performs pretty well. It achieves the highest performance on all the metrics.

**weaknesses:**

1. Some parts of the paper are inconsistent. For example, the author mentioned they used the LDDT score to do the final selection and ranking. However, in experiments, they reported the pLDDT scores instead of LDDT scores.

2. The writing of this paper is bad, which makes this paper hard to follow. For example, when the author merged the VPs into atoms, what information did the author use to identify particles belonging to the same atoms? The 3D coordinates or the atom type? In equation 4, as the LDDT score is a float value, why didn't the author directly use Euclidean distance? After clustering the VPs, $\hat{x}^0$ is randomly sampled from the particles in the cluster while $\hat{y}^0$ is set to the mean position of all particles, which seems not aligned. why didn't the author directly set y to the corresponding coordinate of the sampled x.

3. There are so many hyper-parameters in this paper, such as number of particle movement iteration, the number of initial VPs, VP error threshold, number of molecule refinement iteration, clip value for coordinate loss, threshold for moving regularization... I feel training the model  might be hard.

---

> ### Author Response · Authors · 2023-11-01
> **Response to Reviewer Qyn7**
>
> We sincerely thank the reviewer for their insightful comments and feedback on our paper. We have carefully addressed each concern as detailed below:
>
> 1. **LDDT and pLDDT (Weakness 1)**:
>    - Thanks for the question. pLDDT stands for "predicted LDDT.", which is calculated by the model. Calculating LDDT requires ground-truth labels, which are infeasible to obtain during test time. Therefore, we use pLDDT as a confidence score. We have revised the paper for clarity on this matter.
>
> 2. **Writing and Design Intuition (Weakness 2 and Requested Change 1)**:
>    - We acknowledge that some details and the underlying intuitions behind the model designs were not adequately explained in the paper. In response to the feedback, we've provided a more in-depth explanation and insights in Sec. 2 in the revised version.
>
> 3. **Hyperparameters (Weakness 3)**:
>    - We understand the concern about the number of hyperparameters. In this work, we have introduced several hyperparameters such as loss coefficients, regularizations, $R_1$, $R_2$, and $k_{vp}$ (in Table 4). Among these, $R_1$, $R_2$, and $k_{vp}$ are especially important and were carefully adjusted, as shown in our Ablation Study (Section 3.3). For the other hyperparameters, such as learning rates and dropout rates, we selected values based on conventional wisdom from previous works. Our experiments confirm that our model still outperforms previous baselines, even without tuning these parameters. We have revised the paper to clarify this point.
>
> 4. **Overall Architecture Figure (Requested Change 2)**:
>    - We'd like to clarify that an architectural diagram was present in our initial submission, as shown in Figure 5.
>
> 5. **Evaluation of Diversity (Requested Change 3)**:
>    - Based on the recommendation, we've incorporated an additional experiment focusing on diversity. Kindly refer to Appendix C.4 for further details.
>
> 6. **Additional Questions**:
>
>    - For merging VPs into atoms, the process is primarily based on the 3D coordinates of the atoms, as our objective is to extract the top-m dense positions.
>
>    - In equation 4, we mirror AlphaFold2's approach, converting regression tasks to classification ones. This offers better training stability, and our objective revolves around accurate ranking of pLDDT values, not their precision.
>
>    - Regarding VP clustering, our primary aim is stability, leading us to prefer averaging. While averaging is feasible for 3D coordinates, it isn't for discrete atom types. Hence, the atom types are sampled instead.
>    By sampling repetitively, we achieve an approximation of averaging. With both training and inference processes executed multiple times, we ensure stability in the overall outcome.
>
> Thank you again for your insights, and we hope that our revisions address your concerns.

---

### Review · Reviewer_wMDA · 2023-10-25

**Summary Of Contributions:**

- Proposes VD-Gen, a novel end-to-end pipeline for generating 3D molecules conditioned on protein pockets. The pipeline contains 5 stages: virtual particle initialization, iterative movement, molecule extraction, iterative refinement, and confidence prediction.

- Introduces the concept of using virtual particles to approximate the distribution of molecular atoms within the protein pocket cavity. This allows generating all atoms simultaneously in a non-autoregressive manner, avoiding issues with determining atom generation order.

- Comprehensive experiments demonstrate VD-Gen can generate high quality 3D molecules exhibiting significantly better binding affinities than previous baselines across various metrics. Ablation studies validating the importance of key components like virtual particles, movement iterations, and confidence prediction.

- Case studies and visualizations offer insights into how VD-Gen operates and generates reasonable 3D molecules tailored to pocket structures.

- Extends the pipeline to 3D molecular optimization tasks, still achieving strong performance compared to fragment-based baselines.

- The introduced model offers a promising solution for generating novel drug-like molecules against target pockets, addressing limitations of traditional virtual screening and previous generative approaches.

**Audience:**

Yes

**Broader Impact Concerns:**

- Drug safety/efficacy: While the method aims to improve binding affinity, evaluating full safety and efficacy requires significant further testing. The limitations of computational screening should be clearly stated to avoid overstating capabilities.

- Dual use concerns: While focused on drug discovery, improved generative modeling of chemicals does carry risks of misuse. The authors could affirm their commitment to ethical applications.

- Equitable development: Ensuring equity in who benefits from computational advances in drug development could be considered.

**Claims And Evidence:**

Yes

**Requested Changes:**

- Perform additional experiments on more diverse protein pockets to better validate general applicability. Two are examined in the case study.
- Provide more details on the training data and model architecture to enhance reproducibility. The current model structure and parameter settings are not clearly claimed in the experiment setup, and so are the training data.
- Perform hyperparameter tuning to determine if performance can be further improved. The current experiment can be further  analysed  to explore the method's potential and upper bound.

**Strengths And Weaknesses:**

Strengths:

- The paper addresses an important problem in drug discovery - generating novel 3D molecules that bind to target protein pockets.
The proposed VD-Gen pipeline is novel and represents a clear advancement over previous methods by enabling non-autoregressive generation.
- The virtual particle concept provides an intuitive way to approximate atom distributions for 3D generation.

- The comprehensive set of experiments thoroughly validates the effectiveness of VD-Gen across different metrics.
Ablation studies provide useful insights and validate design decisions like using virtual particles.Extending VD-Gen to optimization tasks demonstrates flexibility of the approach.

Weaknesses:
- Only one dataset was used for evaluation - testing on more diverse pockets could further validate generalizability.
- Hyperparameters were not extensively tuned - optimal settings may further boost performance.
Efficiency and scalability could be better, especially for real-world applications.
- Applicability to very large novel pockets remains uncertain. More rigorous evaluation of generated molecular properties beyond binding affinity could be beneficial

---

> ### Author Response · Authors · 2023-11-01
> **Response to Reviewer wMDA**
>
> We thank the reviewer for the thoughtful feedback. Here are our responses to the concerns raised:
>
> 1. **Diverse Pockets for Evaluation (Weakness 1)**:
>
>     - Thanks for the question. The dataset we used is diverse to validate the generalization ability of our method. We have carefully checked the evaluation samples. There are 100 pockets in our evaluation dataset where MM-PBSA proved effective. To check the diversity of the samples, we calculated the similarity of the pockets with mmseqs and found that at a 40\% similarity threshold, there were over 73 clusters. This clearly shows that our evaluation dataset is diverse. We also ensured generalizability by not using any complexes in the training data that had protein sequences similar to the test set.
>
> 2. **Hyperparameters Tuning (Weakness 2 and Requested Change 3)**:
>
>     - We have conducted tuning for several important hyperparameters; see the ablation study in Section 3.3. We agree that an exhaustive search on the hyperparameter space may further help. However, such a process requires a significant amount of computational cost, which we cannot finish before the rebuttal deadline. Given that our method already performed much better than previous models (which clearly demonstrates the strength of the method), we would like to leave the hyperparameter search results in the next stage.
>
>
> 3. **Evaluation of Generated Molecular Properties (Weakness 3)**:
>
>     - We agree that an in-depth evaluation including factors like ADME/T would be valuable. However, as you've noted in Broader Impact Concerns, evaluating these properties with high accuracy remains a challenge in practice. We've updated our paper to emphasize this fact and to ensure we don't overstate our capabilities.
>
> 4. **More Case Studies (Requested Change 1)**:
>
>     - Based on your suggestion, we've incorporated 3 additional case studies in our paper (Sec. 3.4).
>
> 5. **Training Data and Model Architecture Details (Requested Change 2)**:
>
>     - We used the CrossDocked dataset for training, similar to prior baselines. We've added more descriptions of the dataset in Sec. 3.1 and updated the paper to offer more details to aid reproducibility (after Sec. 5).  Details about our model are in Appendix A, and training settings can be found in Appendix C.1.
>
> 6. **Broader Impact Concerns**:
>
>     - We've incorporated these discussions into the revised paper after Sec. 5.

---

### Comment · Reviewer_XPHM · 2023-11-06
**Response to updated paper**

Thank you very much for responding to our reviews and updating the paper. I have read the reviews from the other reviewers and looked at your responses. Let me try to summarize the points made.

**Contribution**: overall there seems to be an agreement from reviewers that the method is novel and addresses an important problem.

**Hyperparamters**: all reviewers pointed out the large number of hyperparameters in the method as a potential disadvantage. The authors said that the parameters were easy to tune, pointed to their ablation study, and added discussion to section C.1. I am happy with this. Reviewer wMDA asked for an additional tuning study: I personally think this is unnecessary.

**Evaluation**: reviewers wMDA and Qyn7 asked for extra evaluation (e.g. quality of generated molecules, diversity, larger pockets). I personally think this is not necessary, but don't work on conformer generation myself so am not confident about this.

**Clarity of the method**: I agree with reviewer Qyn7 that the method can be explained better. I think the authors have made some improvements but that things are not totally clear. In particular:
- In section 2, $\mathcal{M}$ is not clearly defined: is this the space of molecules, or a distribution over molecules? More generally, is $h$ a training procedure that outputs a distribution over molecules, or does it define a distribution over molecules. The paper seems to imply the former, while the latter is generally what is called a "generative model" in machine learning.
- More generally, I think the authors have misused $\mapsto$: I think it refers to the *space* which the function outputs to, *not* the variable that receives this output

**Speculation/unsupported claims**: the authors responded to many reviewer points with what seems like [unfounded] speculation. In particular, the following claims stand out:

> Diffusion models, using a small-step denoising technique, require many more steps (e.g., 1000) to produce satisfactory results and don't consider the optimal transport, which might affect their performance negatively.

^ Firstly, it is not clear that they *require* many more steps (previous work just used this). Secondly, the remark about optimal transport seems purely speculative.

> VD-Gen, in contrast, uses a many-to-1 mapping to improve optimization locality, ensuring a stable training process. Despite the need for a more complex pipeline to merge the additional points, Table 7's results show VD-Gen surpassing diffusion models. Moreover, VD-Gen's performance in Table 9 shows it is approximately ten times faster than diffusion models.

^ You appear to test against only one diffusion model, which I think is insufficient evidence to claim that VD-Gen is _generally_ faster than diffusion models

> [taken from their response to me] VD-Gen, 3DSBDD, and Pocket2Mol inherently rank and select generated molecules, minimizing randomness. For instance, 3DSBDD and Pocket2Mol use beam-search for consistent results, while VD-Gen utilizes predicted confidence scores for top molecule selection.

> Each method selects 100 molecules from a 500-molecule pool per pocket, as detailed in Section 3.2. Table 1 reports the 5th percentile results, with other percentiles in Appendix C.3. Given these metrics reflect aggregate results, they sufficiently account for any randomness.

^ This is pure speculation about why the variation *might* be low, and excludes any possible variation from training (i.e. two training runs may yield different model parameters $\theta$). I do not think that claims of superior performance can be made without error bars!

**Overall**: I imagine that the reviewers will reach consensus to accept this paper, but it would be nice to see some further improvements made by the authors.

---

> ### Author Response · Authors · 2023-11-07
> **Thank you very much for the further comments**
>
> We are grateful for the comprehensive summary of the review comments, the supportive feedback on our work, and the additional comments provided. We have further revised our manuscript in response to these further comments as outlined below.
>
> 1. **Notations**
>
>    Thanks for the suggestions. We have updated the notations in Section 2 for a better presentation. And we use symbol $\to$ to represent the space mapping of a function, as referenced in [1, 2].
>
> 2. Regarding **unsupported claims**
>
>    We apologize for any unsupported assertions made in our previous draft. The paper has been revised to ensure our claims about general diffusion models are accurate and fully substantiated.
>
> 3. Regarding **error bars**
>
>    We acknowledge the importance of including error bars and will incorporate them into our results. However, we would like to point out that previous studies included **no** error bars, and they only released one model checkpoint, making it hard to compute the "standard deviations". We can try to reproduce the training of baselines with different seeds. However, due to time constraints and the substantial effort required, we may not be able to complete all replication experiments before the deadline.
>
>
> Reference:
>
> [1] https://en.wikipedia.org/wiki/Glossary_of_mathematical_symbols
>
> [2] https://math.stackexchange.com/questions/936558/use-of-mapsto-and-to

---

> > ### Comment · Reviewer_XPHM · 2023-11-08
> > **Happy with changes**
> >
> > Thank you for making further updates to the manuscript. I think:
> >
> > - The notation has improved and I have no further comments
> > - I am happy with the revised claims
> > - I am aware that previous studies did not provide error bars. This is bad science and if I were a reviewer of those works I would have also asked for error bars. I think it is enough for you to provide error bars for your own method (VD-Gen), but not for the baseline methods. With regards to completing things before a "deadline", TMLR has no deadlines so it should be fine. I know they aim to complete the reviewing within a short period, but you are certainly allowed to request extra time (I have seen this happen with other submissions).

---

> > > ### Author Response · Authors · 2023-11-13
> > > **Response to Reviewer XPHM**
> > >
> > > We have trained two additional VD-Gen models with different seeds,  computed the mean and standard deviation of the inference results, and revised the experimental tables accordingly. Notably, the small error bars in the results indicate that VD-Gen consistently surpasses previous models, even when considering for randomness.  We greatly appreciate your valuable suggestions!

---

> > > > ### Comment · Reviewer_XPHM · 2023-11-19
> > > > **Good changes but you should label the +/- values**
> > > >
> > > > Thank you for this change. Maybe add an explanation in the caption of the Table stating that the $\pm$ values are standard deviations and not e.g. standard errors? At the moment this seems unclear.

---

> > > > > ### Author Response · Authors · 2023-11-19
> > > > > **Response to Reviewer XPHM**
> > > > >
> > > > > We apologize for any confusion. The table captions have been updated for clearer explanations.

---

### Comment · Action_Editors · 2024-02-18

The paper present a new method for molecule generation using 3D particles of a binding pocket.  All reviewers agree with the novel contribution of paper, the technique of using virtual particles to approximate atom distribution, and the strong experimental results.
Reviewers have concerns about the complexity of the method -- too many hyper-parameters. Some parts of the paper could be revised with a clearer description (e.g. section 2), the claim about diffusion models, and the analysis on variance.
The author may want to revise according the detailed comments of three reviewers.
Overall, the paper made adequate contribution and could be included in TMLR with small modification.
I  recommend Accept with minor revision.

---

### Decision · Action_Editor_EbNb · 2024-03-02

**Recommendation:** Accept with minor revision

**Comment:**

All reviewers agree with the novel contribution of paper, the technique of using virtual particles to approximate atom distribution, and the strong experimental results.

Reviewers have concerns about the complexity of the method -- too many hyper-parameters. Some parts of the paper could be revised with a clearer description (e.g. section 2), the claim about diffusion models, and the analysis on variance.

The author may want to revise according the detailed comments of three reviewers.

Overall, the paper made adequate contribution and could be included in TMLR with small modification. I recommend Accept with minor revision.

**Audience:**

It will be relevant researchers working on molecule generation.

**Claims And Evidence:**

The paper present a new method for molecule generation using 3D particles of a binding pocket. The claims are supported by experiments.

---

> ### Author Response · Authors · 2024-03-12
> **Minor Revisions**
>
> Thank you for your insightful feedback. We are both thrilled and honored to receive your recommendation.
> We have incorporated the suggested minor revisions in our updated camera-ready version. More Specifically, we clarified hyperparameters tuning and added more details about training.  And we revised the writing with more details and clarity, including clear notations, more case studies and more details for reproducibility. And we added an in-depth discussion on diffusion models, as well as the variance analysis for VD-Gen's results.